# Graph-based Unsupervised Disentangled Representation Learning via Multimodal Large Language Models

**Baao Xie[1]  Qiuyu Chen[1,2]  Yunnan Wang[1,2]  Zequn Zhang[1,3]  Xin Jin[1,*]  Wenjun Zeng[1]**

[1]Ningbo Institute of Digital Twin, Eastern Institute of Technology, Ningbo, China
[2]Shanghai Jiao Tong University, Shanghai, China
[3] University of Science and Technology of China, Hefei, China
bxie@idt.eitech.edu.cn   jinxin@eitech.edu.cn

## Abstract

Disentangled representation learning (DRL) aims to identify and decompose underlying factors behind observations, thus facilitating data perception and generation. However, current DRL approaches often rely on the unrealistic assumption that semantic factors are statistically independent. In reality, these factors may exhibit correlations, which off-the-shelf solutions have yet to properly address. To tackle this challenge, we introduce a bidirectional weighted graph-based framework, to learn factorized attributes and their interrelations within complex data. Specifically, we propose a $\beta$-VAE based module to extract factors as the initial nodes of the graph, and leverage the multimodal large language model (MLLM) to discover and rank latent correlations, thereby updating the weighted edges. By integrating these complementary modules, our model successfully achieves fine-grained, practical and unsupervised disentanglement. Experiments demonstrate our method's superior performance in disentanglement and reconstruction. Furthermore, the model inherits enhanced interpretability and generalizability from MLLMs.

## 1  Introduction

Disentangled representation learning (DRL) is a major goal of artificial intelligence (AI), acclaimed for its enhancement of model robustness, interpretability, and generalizability. Essentially, DRL methods imitate the understanding processes of biological intelligence, wherein comprehension of real-world is achieved by separating observations into distinct factors [1]. In this form, specific attributes (e.g., object color, shape, and size) exhibit exclusive sensitivity to the changes of specific factors. Learning of such disentangled representations is of great importance across various domains, e.g., computer vision [2, 3, 4, 5, 6], natural language processing [7, 8, 9], and AI generated content [10, 11, 12, 13]. In the current phase, unsupervised DRL methods primarily utilize the Variational Autoencoder (VAE) framework [14], a probabilistic model learning representations through a regularization term. This term involves the Kullback-Leibler divergence between the posterior distribution of latent factors and a standard multivariate Gaussian prior, thereby encouraging the factorized representations. To strengthen disentanglement, co-current research [15, 16, 17, 18] focus on the optimization and refinement of the original VAE regularizers, resulting in the family of VAE-based DRL approaches.

Despite the advanced results of the simple and synthetic datasets, VAE-based DRL methods still fall short in interpretability and robustness that are required for effective disentanglement in complex data [19]. This limitation mainly stems from the unrealistic assumption that underlying factors

---

*Xin Jin is the corresponding author. Code is available at here.

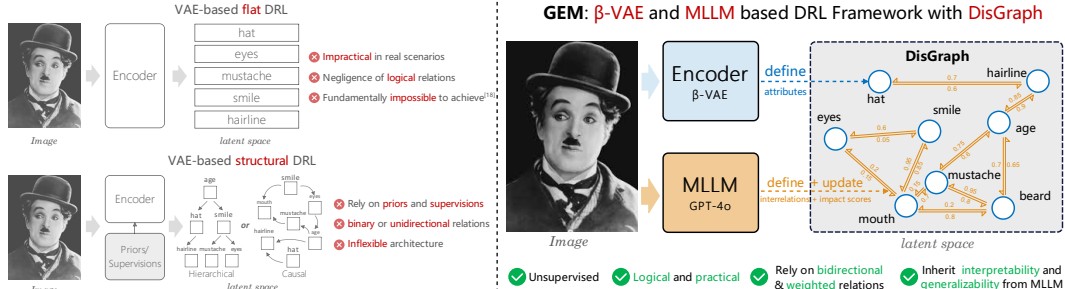

Figure 1: The comparison of typical DRL frameworks with our GEM. The limitations of conventional DRL methods are presented on the left. Conversely, the right-hand side illustrates the advantages of our framework, which benefited from the integration of the $\beta$-VAE and MLLMs.

are countable, independent, and can be fully disentangled in an unsupervised manner (refer to the top-left of Figure 1). In contrast, the real-world variables are pervasively correlated: red apples are more common than yellow ones; elderly people are more frequent with white hair and a receding hairline. Accordingly, an increasing number of recent studies [20, 21, 19, 22] showcases that purely unsupervised DRL is fundamentally impossible without extra priors and inductive biases.

The struggle of typical DRL methods on the complex data returns us back to the essential goal of DRL, i.e., understanding the world as biological intelligence does. This cognitive process can be naturally segmented into three phases: attribute extraction, interrelation perception, and knowledge combination [23, 24], where the latter two stages should not be neglected. From this perspective, several structured DRL approaches, typically known as Hierarchical DRL [25, 26, 12] and Causal DRL [21, 27, 28, 29], have involved the correlations between attributes. However, these approaches usually require extra supervision, and their relations are invariably represented by binary and unidirectional fusion, thus limiting the model performance in practical scenarios (refer to the bottom-left of Figure 1). Inspired by the analysis above, we argue that an effective and practical disentanglement framework should meet the following criteria: (i) the framework should be fully unsupervised; (ii) the framework should be able to disentangle factors while concurrently discovering logical interrelations among them; (iii) the interrelations should be modeled as bidirectional, with corresponding impact scores assigned to each, thereby improving model performance in complex scenarios. On this basis, we propose a novel **G**raph-based dis**E**ntanglement framework with **M**ultimodel large language models, dubbed **GEM**. Specifically, our model employs two complementary branches: a $\beta$-VAE based disentanglement branch for the attribute extraction, and a multimodal large language model (MLLM) based branch for the interrelation discovery. The relation-aware representations are further embedded into a disentangled bidirectional weighted graph (DisGraph), which presents distinct factors as nodes, interrelations as edges, and impact scores as weights. The parameters of the graph are dynamically updated and refined via a graph learner. The experimental results show that GEM achieves superior performance on fine-grained and relation-aware disentanglement, while preserving the reconstruction quality. Furthermore, the model is endowed with superior interpretability and generalizability that derived from MLLMs. All in all, our main contributions can be summarised as:

- To our best knowledge, we are the first to leverage the commonsense reasoning of MLLMs to discover and rank the semantic interrelations from the perspective of DRL.

- We propose a novel and practical disentanglement framework built upon $\beta$-VAE and MLLMs to learn the independent factors and their interrelations in an unsupervised way.

- We introduce a bidirectional and self-driven graph architecture to encode the relation-aware representations, thus facilitating practical and controllable disentanglement.

## 2 Related Work

### 2.1 Standard Disentangled Representation Learning

The definition of DRL is intuitively given by Bengio et al. [1] as a technique to separate semantic factors behind observational data. This approach assumes that individual data attributes are sensitive to changes in single latent factors, while not being affected by other factors. The disentanglement of

attributes is believed helpful for downstream tasks, e.g., generative models [3, 30, 5, 31, 32], medical imaging [33, 34, 35], image editing [36, 37, 38, 39], and 3D reconstruction [40, 41, 42].

Traditional DRL methods primarily utilizes the VAE framework, achieving a measure of disentanglement on static datasets. This framework has been further enhanced by extensive models such as $\beta$-VAE [15], $\beta$-TCVAE [16], DIP-VAE [4], FactorVAE [43], RF-VAE [44], and $\alpha$-TCVAE [18] through the optimizations of regularization terms. Despite the successes on simple and static datasets, standard DRL approaches still encounter challenges in complex data. It is mainly due to the flat and unrealistic assumption: data properties are independent and can be factorized into distinct factors [1, 21, 45, 46]. Locatello et al. [20] have proven that unsupervised DRL is fundamentally impossible without extra priors. Thus, subsequent studies have demonstrated that a practical DRL model with appropriate inductive biases can enhance the disentanglement in real scenes [47, 48, 49, 50].

## 2.2 Structured Disentangled Representation Learning

In contrast to the flat and VAE-based DRL methods, recent research gradually realize that latent factors might naturally involve semantic interrelations, deriving to the branch of structured disentangled representation learning [19]. Within this domain, Hierarchical DRL and Causal DRL are mostly relevant to our work. Hierarchical DRL presumes that underlying factors have different levels of semantic abstraction, either dependent [51] or independent [25] across levels. While straightforward, Li et al. [25] propose a hierarchical VAE-based model to learn semantic representations. Furthermore, Singh et al. [12] introduce FineGAN, a three-tier hierarchical framework for controllable object generation. Li et al. [52] also propose a hierarchical DRL framework aimed at facilitating image-to-image translation. Differently, our framework aims to achieve fine-grained disentanglement, where the targeted attributes are always flat, e.g., the wrinkle, lipstick, and mustache of faces. Therefore, we rely on the flat representations, but place a strong emphasis on the mutual relations between attributes.

Similarly, Causal DRL methods endeavor to capture the causal relations between disentangled factors. As the first, Yang et al. [21] propose CausalVAE to discover relations from the perspective of causality. Further, Shen et al. [29] propose a weakly supervised framework DEAR with the structured causal model (SCM) as prior. However in our view, current Causal DRL methods have at least three unpractical issues: (i) rely on various degrees of supervision; (ii) aim to model a specific event rather than a common scenario; (iii) the causal relationship is often overly simplistic, being impractically binary and unidirectional, i.e., paired variables A and B only have two possible causal relations: either $A \rightarrow B$ or $A \leftarrow B$ (otherwise unrelated). In practical, it is common for paired variables to exhibit bidirectional influence, and the impact of such bidirectional relations should be properly ranked.

## 2.3 Multimodal Large Language Models

Recent years have witnessed the remarkable advancements in Multimodal Large Language Model (MLLM) [53, 54, 55, 56]. Since the release of Generative Pre-trained Transformer (GPT) [57], there has been a research trend over MLLMs regarding to its demonstrated potential in processing multimodal data [58, 59, 60]. As the variants of GPT-4, GPT-4 with Vision (GPT-4V) [61] and GPT-4 omni (GPT-4o) [62] enhance to process textual and visual data, enabling richer, context-aware interactions across a range of multimodalities. Concurrently, following works such as Gemini [58], Claude [63], NExT-GPTs [64] and GLM-4 [65] have strengthen the support to additional modalities.

The powerful capacities of MLLMs gradually make researchers aware of its latent perceptual knowledge embedded within networks. Gandelsman et al. [59] investigate the way that CLIP encoder understands visual data, by decomposing representations into individual components. In addition, Basu et al. [66] propose a mechanistic localization approach to explore how the visual properties are encoded in MLLMs. However, to our best knowledge, there is limited exploration into leveraging the commonsense reasoning of MLLMs from the perspective of DRL. And we are the first to employ MLLMs to discover and rank interrelations between semantic factors in the DRL framework.

## 3 Methodology

To achieve fine-grained and relation-aware disentanglement, we propose GEM, a novel and practical framework that synergizes the strengths of DRL and MLLMs by a bidirectional weighted DisGraph. As depicted in Figure 2, GEM is comprised of two complementary modules: a $\beta$-VAE based

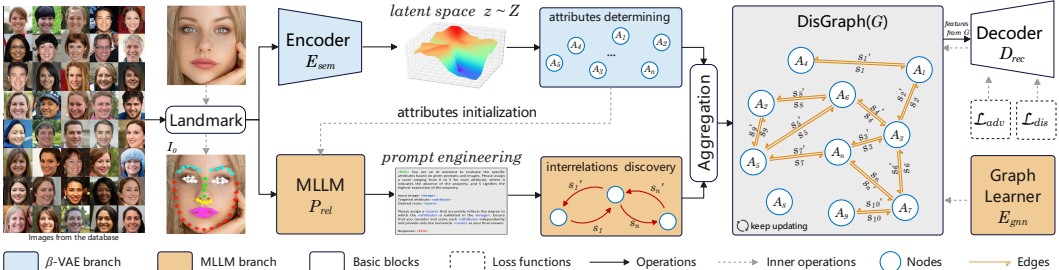

Figure 2: Pipeline of our GEM. The model consists of two complementary branches, termed as a $\beta$-VAE branch (blue) and a MLLM branch (brown). The former utilizes $\beta$-VAE based semantic encoder $E_{sem}$ to disentangle underlying factors, while the latter employs prompt engineering to discover and rank interrelations. The bidirectional weighted DisGragh $G$ is further proposed to embed relation-aware representations, with its parameters optimized constantly by a GNN network $E_{gnn}$.

branch dedicated to extract attributes (Section 3.1), and a MLLM-based branch to discover and rank interrelations (Section 3.2). The relation-aware representations are then embedded into the DisGragh (Section 3.3), which presents factors as nodes, interrelations as edges, and impact scores as weights.

## 3.1 $\beta$-VAE based Attribute Determining Branch

The fundamental objective of vanilla VAE is to approximate data distributions by employing a maximum likelihood estimation framework as outlined in Eq. 1:

$$\log p_\theta(\mathbf{x}) = D_{KL}\left(q_\phi(\mathbf{z}|\mathbf{x})\|p_\theta(\mathbf{z}|\mathbf{x})\right) + \mathcal{L}(\theta, \phi), \tag{1}$$

where the variational posterior distribution $q_\phi(\mathbf{z}|\mathbf{x})$ is utilized to represent the probability distribution of the latent variable $z$ given the observation $x$. The key of Eq. 1 is maximizing the approximation $\log p_\theta(\mathbf{x})$ of the true posterior distribution $p_\phi(\mathbf{z}|\mathbf{x})$ and $q_\phi(\mathbf{z}|\mathbf{x})$.

Specifically, the first term of Eq. 2 corresponds to the Kullback-Leibler (KL) divergence measuring the distance between distribution $q_\phi(\mathbf{z}|\mathbf{x})$ and $p_\theta(\mathbf{z}|\mathbf{x})$. The second term is denoted as the variational evidence lower bound (ELBO). Empirically, the maximization of ELBO is employed to provide a stringent tight lower bound for the original log-likelihood $\log p_\theta(\mathbf{x})$. ELBO can be reformulated as:

$$\mathcal{L}(\theta, \phi) = \mathbb{E}_{q_\phi(\mathbf{z}|\mathbf{x})}\left[\log p_\theta(\mathbf{x}|\mathbf{z})\right] - \beta D_{KL}\left(q_\phi(\mathbf{z}|\mathbf{x})\|p_\theta(\mathbf{z})\right), \tag{2}$$

where the initial term, i.e., conditional logarithmic likelihood $\mathbb{E}_{q_\phi(\mathbf{z}|\mathbf{x})}\left[\log p_\theta(\mathbf{x}|\mathbf{z})\right]$ is responsible for the reconstruction. Typically, the latent variable $z$ is assumed to follow a standard Gaussian distribution $\mathcal{N}(0, 1)$ for $p_\theta(z)$, so that the KL term actually imposes independent constraints on the representations. Furthermore, subsequent studies [15, 17, 67] highlight that a extra penalty coefficient prior to the KL term, denoted by $\beta$, can significantly strengthen disentanglement. When $\beta$ is set to 1, the $\beta$-VAE reverts to the standard VAE framework. And an increase in $\beta$ encourages more disentangled representations but harms the performance of reconstruction as a trade-off. As per the Information Bottleneck (IB) theory [17], constraining the information input to DRL models (e.g., via $\beta$ penalty coefficient) inherently enables them to identify and learn the most representative factors for successful reconstruction. For instance, when trained on the Shapes3D (a collection of synthetic objects) with a merely three-dimensional latent variable, the attribute determining branch tends to learn the most critical factors, observed to be "object color", "object shape", and "background shape". These attributes are organized in the three dimensions, ordered by their reconstruction contribution.

Specifically, within the processes of this branch, the input image is firstly subjected to a pre-processing step utilizing landmark detection functions as instructed by [68] and [69] (see Figure 2). It serves as a regularization phase, to remain the key features through targeted cropping. Additional derivations of this process are documented in the appendix. Then, the pre-processed $I_0$ is fed into a $\beta$-VAE based branch, designed to disentangle factors associated with each dimension in the latent variable $z \in Z$. However, the input of decoder $D_{rec}$ is the relation-aware variable $z_{rel} = \mathbf{A}^T z$ from the DisGraph, rather than the $z \in \mathcal{Z}$. It means the prior assumption of $p_\theta(z) \in \mathcal{N}(0, 1)$ is no longer hold. To address this issue, we reformulate the loss function in $\beta$-VAE as follows:

$$L_{gem}(\phi, \gamma, \theta) = D_{\mathrm{KL}}(q_\phi(x, z), p_{\gamma, \theta}(x, z)) \tag{3}$$

$$\nabla_\theta L_{gem}(\phi, \gamma, \theta) \overset{x=D_\theta(z)}{=} -E_{z \sim q(z)} \nabla_x \left[\log\left(\frac{p_{\theta, \gamma}(x, z)}{q_\phi(x, z)}\right)\right] \nabla_\theta x \tag{4}$$

$$\nabla_\phi L_{gem}(\phi, \gamma, \theta) \stackrel{z=E_\phi(x)}{=} E_{x \sim p(x)} \nabla_z \left[ \log \left( \frac{p_{\theta,\gamma}(x,z)}{q_\phi(x,z)} \right) \right] \nabla_\phi z \qquad (5)$$

$$\nabla_\gamma L_{gem}(\phi, \gamma, \theta) \stackrel{z=G_\gamma(z)}{=} E_{x \sim p(x)} \nabla_z \left[ \log \left( \frac{p_{\theta,\gamma}(x,z)}{q_\phi(x,z)} \right) \right] \nabla_\gamma z \qquad (6)$$

where the $\phi$, $\theta$ and $\gamma$ are the learnable parameters of $E_{sem}$, $D_{rec}$ and DisGraph $G$, respectively. Let's say $D(x,y) = \log \left( \frac{p_\theta(x,z)}{\beta q_\phi(x,z)} \right)$, and the the gradients with respect to $x$ and $z$ can be obtained during backpropagation by the cross-entropy:

$$\mathcal{L}_{adv} = \mathcal{L}_{D(x,y)} = \frac{1}{N_{bc}N_m} \left[ \sum_{i=0}^{N_{bc}} \text{softplus} \left( -D\left(x_i, z_i\right) \right) + \sum_{i=0}^{N_{bc}} \text{softplus} \left( D\left(x_i, z_i\right) \right) \right] \qquad (7)$$

where $N_{bc}$ and $N_m$ represent the number of samples and the posterior samples in a batch, respectively. Obviously, this loss resembles the adversarial loss utilized in Generative Adversarial Networks (GAN) [70]. Therefore, we employ the adversarial training strategy to optimize $D(x,y)$. Combined with the disentanglement term from the original $\beta$-VAE indicated as $\mathcal{L}_{dis}$, the total loss for the attribute determining branch can be expressed as:

$$\mathcal{L}_{total} = \lambda_{adv}\mathcal{L}_{adv} + \lambda_{dis}\mathcal{L}_{dis} + \lambda_{gem}\mathcal{L}_{gem} \qquad (8)$$

where the $\lambda_{adv}$, $\lambda_{dis}$ and $\lambda_{gem}$ serve as hyperparameters to balance the disentanglement capability and reconstruction quality, with default values set to 0.8, 0.6 and 0.6, respectively. The detailed derivation process of the adversarial training strategy is provided in the supplementary material.

### 3.2 MLLM-based Interrelation Discovery Branch

Given a pre-processed image $I_0$ with $n$ targeted attributes $\mathcal{A} = \{1, 2, 3, ..., n\}$ initialized by the $\beta$-VAE branch, our objective is to discover and rank the mutual relations for each pair within $\mathcal{A}$. As represented by the brown blocks in Figure 2, we employ MLLMs as a relation predictor $P_{rel}$ to discover and rank interrelations. Initially, the MLLM is required to score from 0 to 5 for each attribute, where 0 indicates the attribute's absence, and 5 denotes its highest expression. As shown in Figure 3, the queries can be formulated as a question in natural language with the input image $I_0$.

Based on the attribute scores, we subsequently employ Somers' D algorithm [71] to rank the bidirectional impact scores of interrelations. For the attribute pair $(A_i, A_j)$, we determine the number of concordant pairs $N_C$ and discordant pairs $N_D$, as delineated by Kendall's Tau [72] algorithm. Subsequently, the impact score $\mathcal{S}_{ij}$ within $\mathcal{S} = \{1, 2, 3, ..., k\}$ can be denoted as:

$$\mathcal{S}_{ij} = \frac{N_c - N_d}{N_c + N_d + T_i} \qquad (9)$$

For the reversed relation of $(A_i, A_j)$, the impact score can be denoted as $\mathcal{S}_{ji}$ or $\mathcal{S}'_{ij}$:

$$\mathcal{S}'_{ij} = \mathcal{S}_{ji} = \frac{N_c - N_d}{N_c + N_d + T_j} \qquad (10)$$

where $T_i$ and $T_j$ is the number of ties only for the independent variable $A_i$ and $A_j$, respectively. The calculated $\mathcal{S}$ and $\mathcal{S}'$ are used for initialization and refinement of DisGraph (see Section 3.3). As illustrated in Figure 4, it is important to clarify that the primary goal of the MLLM branch in GEM is to discover interrelations, where the statistical relativity between two attributes is of primary concern, rather than the absolute scores for the individual attribute. For example, given a collection of facial images, it is acceptable if the scores of "age" and "bald" exhibit a positive correlation, even if the specific score values are fluctuating. To ascertain the reliability of MLLMs for interrelation discovery, extra experiments are performed as shown in Section 4.4.

> <BOS> You are an AI assistant to evaluate the specific attributes based on given prompts and images. Please assign a score ranging from 0 to 5 for each attribute, where 0 indicates the absence of the attribute, and 5 signifies the highest expression of the attribute.
>
> Input image: <image>
> Targeted attribute: <attribute>
> Desired degree score: <score>
>
> ........................................................................
>
> Please assign a <score> that accurately reflects the expressive level to which the <attribute> is exhibited in the <image>. Ensure that you consider and score each <attribute> independently and provide only the numerical <score> score as your final answer.
>
> Response: <EOS>

Figure 3: A simplified example of the template for prompting MLLMs to evaluate attributes. Specifically, <text> is the interactive token, while <BOS> and <EOS> are tokens denoting the start and end of the input to MLLMs, respectively.

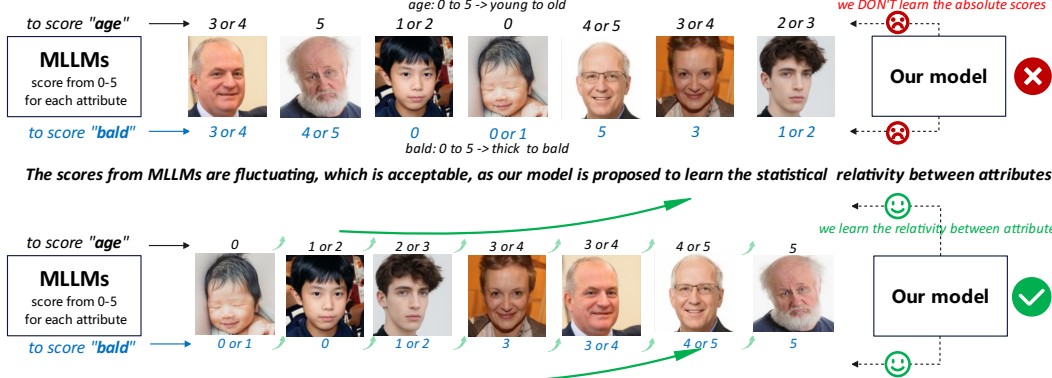

Figure 4: Our aim is using the commonsense knowledge behind MLLMs to equip GEM with ability of interrelations discovery, where a certain degree of fluctuations on absolute scores are acceptable.

## 3.3 Bidirectional Weighted DisGraph

Based on the extracted factors and interrelations, we then propose the bidirectional weighted DisGraph $\mathcal{G} = (\mathcal{A}, \mathcal{E}, \mathcal{S})$ to integrate the semantic representations. Specifically, $\mathcal{A}$ is the set of $n = |\mathcal{A}|$ nodes, embodying the disentangled attributes as factors. Besides, $\mathcal{E}$ is the set of $k = |\mathcal{E}|$ edges, and $\mathcal{S}$ stands for the weights of these edges. An $e \in \mathcal{E}$ and its corresponding impact score $s \in \mathcal{S}$ are embedded. Consequently, $\mathcal{G}$ can be presented as the learnable weighted adjacency matrix $\mathbf{A} \in [0, 1]^{n \times n}$.

According to the definitions above, the model firstly constructs a sketched adjacency matrix $\mathbf{A}_0 \in \mathbb{R}^{n \times n}$ upon the factors and relations initialized by the $\beta$-VAE branch and MLLM branch. Specifically, we treat the averaged impact scores of the first 1,500 images processed by MLLMs, as initial weights of relations. We further employ an unsupervised graph learner $E_{gnn}$ to dynamically refine the parameters of DisGraph by the structure bootstrapping mechanism [73] and multi-view graph contrastive learning [74]. The optimization function of $E_{gnn}$ can be formulated as:

$$\mathbf{T}^{(l)} = h_w^{(l)}\left(\mathbf{T}^{(l-1)}, \mathbf{A}\right) = \sigma\left(\widetilde{\mathbf{D}}^{-\frac{1}{2}} \widetilde{\mathbf{A}} \widetilde{\mathbf{D}}^{-\frac{1}{2}} \mathbf{T}^{(l-1)} \Omega^{(l)}\right), \tag{11}$$

It converts the sketched adjacency matrix $\mathbf{A}_0$ into node embedding $\mathbf{T}$ via the GNN-based multilayer network, where $h_w^{(l)}(\cdot)$ is the embedding function with learnable parameters $w$ of the $l$-th layer and $\mathbf{T}^{(l)}$ is the output matrix. The augmented adjacency matrix $\widetilde{\mathbf{A}} = \mathbf{A} + \mathbf{I}$ incorporates self-loops based on the initial matrix $\mathbf{A}_0$, and $\widetilde{\mathbf{D}}$ is the degree matrix of $\mathbf{A}$. Further, $w^{(l)} = \Omega^{(l)} \in \mathbb{R}^{n \times n}$ denotes the parameter matrix of the $l$-th layer, with $\sigma(\cdot)$ as a non-linear function that enhances training stability.

Figure 5 illustrates the comprehensive training algorithm of our model. The encoder processes input images and outputs the disentangled latent variable $z$, which subsequently initializes the embeddings of nodes in Dis-Graph. The adjacency matrix of DisGraph is calculated using Somer's D algorithm, which processes the attribute scores outputted by the MLLM. Following this initialization, a Graph Neural Network (GNN) refines the structure of DisGraph. The average of the feature matrix within DisGraph is then forwarded to the decoder to reconstruct images.

---
**Algorithm 1:** Training Algorithm of GEM

**Input:** Image dataset $X$, Encoder $E_{sem}$, DisGraph $G$, Decoder $D_{rec}$, Discriminator $D$, parameters of the encoder $E_{sem}$, DisGraph $G$ decoder $D_{rec}$ and Discriminator $D$ are denoted as $\phi$, $\gamma$, $\theta$ and $\alpha$

**Output:** Disentangled latent variable $z \in \mathbb{R}^{pre}$, Correlation-involved latent variable $z_{rel} \in \mathbb{R}^{pre}$, reconstructed image $\hat{x}$

```
1  while i ≤ N do
2  │   att_i = MLLM(x_i);              // Getting the attribute scores by MLLM
3  end
4  A_adj ← SomersD(att);    // Getting adjacency matrix by Somer's D algorithm
5  while i ≤ T do
6  │   z = E_sem(x_i);
7  │   embeddings[i] ← Mask(z,i);
8  │   embeddings ← G(A_adj, embeddings);
   │   // Refine the DisGraph by using GNN
9  │   z_rel ← (1/N_node)Σ_i embeddings[i] ;
10 │   x̂ = D(z_rel) ;
11 │   Calculate L_gem, L_dis and L_adv;
12 │   θ ← θ − η_1∇_θL_gem(φ,γ,θ) − η_2∇_θL_dis(φ,θ);
13 │   φ ← φ − η_1∇_φL_gem(φ,γ,θ) − η_2∇_φL_dis(φ,θ);
14 │   γ ← γ − η_3∇_γL_gem(φ,γ,θ);
15 │   α ← α − η_4∇_αL_adv(α) ;
16 │   i = i + 1;
17 end
```

Figure 5: Overall training algorithm of GEM.

Concurrently, the discriminator is trained to approximate the gradient of the loss function. Assuming that the model's performance is upper bounded by the norm of its gradient, which satisfies the Polyak-Lojasiewicz (PL) condition, this configuration ensures the suboptimality of the model.

# 4 Experiments

**Datasets.** We evaluate the GEM on two datasets: 1) **CelebA** [75] contains over 200,000 high-quality facial images. Each image is annotated with 40 binary attribute labels, making it a widely used benchmark for supervised DRL methodologies. Operating in an unsupervised manner, we do not utilize ground-truth labels from this dataset, yet we still conduct comparisons against the supervised approaches; 2) **LSUN** [76] consists of about one million images across various object categories such as cars, buildings, animals, etc. We select a typical subset from both scene categories and object categories, as bedroom and horse, respectively. We believe these two datasets are diverse enough to assess our method covering complex data of different object types.

**Implementation details.** We implement GEM with PyTorch [77]. The landmark pre-processing settings follow the instructions of [68] and [69]. In addition, we employ the latest GPT-4o [62] as the interrelation predictor. For every experiment, the latent dimension size is set to 6. Concentrating on the disentanglement capacity of the framework, all experimental images are resized to a resolution of 64×64 to minimize computational resources. For high-definition outcomes at 256×256, refer to Appendix A.7. All the experiments are processed using the Adam optimizer with a learning rate of 1e-4, and conducted on the Nvidia Tesla A100 GPUs, with a batch size of 32.

**Baselines for Comparison.** We evaluate the GEM with state-of-the-art DRL methods on the disentanglement capacity, reconstruction quality, and computational efficiency. The comparison encompasses supervised and unsupervised models, including standard VAE [14], $\beta$-VAE [15], $\beta$-TCVAE [16], FactorVAE [43] and DEAR [29]. All baselines are trained using the complete CelebA dataset under the configurations previously specified.

## 4.1 Qualitative Results

To evaluate the GEM's effectiveness of relation-aware and fine-grained disentanglement, we perform qualitative analyses with FactorVAE [43] and DEAR [29]. The experiments are conducted on CelebA, a standard benchmark that has been previously validated as compatible with these methods. We select the six fine-grained facial attributes from the database including *Bangs*, *Bald*, *Gender*, *Beard*, *Blond*, and *Makeup*. The disentanglement results are represented by traversals across various latent dimensions, where each dimension corresponds to distinct attributes.

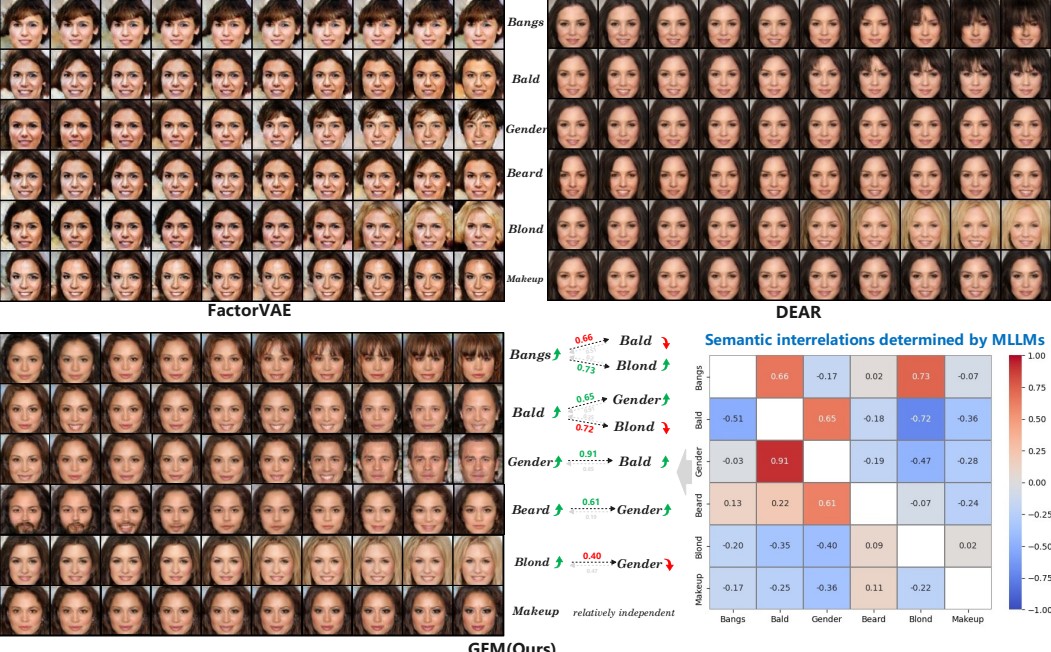

Figure 6: Qualitative comparisons between GEM and typical DRL Methods. Each row in facial images corresponds to the traversal results on a specific attribute, as indicated adjacent to the images (i.e. *Bangs*, *Bald*, *Gender*, *Beard*, *Blond*, and *Makeup*). GEM exhibits superior ability in fine-grained disentanglement with discovered practical and bidirectional relations (illustrated by the heatmap).

As illustrated in Figure 6, GEM effectively achieves fine-grained and relation-aware disentanglement, via the integration of DRL and MLLMs. The interrelations determined by MLLMs are depicted as a heatmap in the bottom-right of Figure 6, where deeper colors reflect stronger relations. Since DisGraph is bidirectional, the impact scores for bidirectional relations between a pair of attributes may vary, resulting in an asymmetric matrix. Specifically, in the first row of GEM's result, a person with heavier *Bangs* is less likely to be *Bald*, and the hair tends to be *Blond*, which is considered logical by MLLMs. Furthermore, as shown in the second and third rows, males (*Gender*) are more likely to be *Bald* and less likely to have *Blond* hair. The attribute *Makeup* is considered as relatively independent, with lower impacts scores among other attributes.

In comparison, DEAR demonstrates limitations in learning specific attributes such as *Bald* (second row) and *Gender* (third row), while the relations between attributes appear to be tenuous. To our knowledge, this underperformance may stem from the stringent nature of causal relations, which are single-directional and heavily rely on the quality of prior. For FactorVAE, since it is a flat DRL framework, we employ the same causal relations used in DEAR to make it relation-aware. As shown in Figure 4, GEM surpasses FactorVAE in both attribute disentanglement and relation discovery, which indicates the importance of specially-designed modules within our framework.

## 4.2 Quantitative Results

Table 1 reports the results of Frechet Inception Distance (FID) [70] and Kernal Inception Distance (KID) [70] scores to verify the quality of reconstructed images. To ensure statistical significance, each comparison model undergoes three rounds of evaluations in the same configuration. The results indicate that GEM outperforms both typical unsupervised (VAE, $\beta$-VAE, $\beta$-TCVAE, FactorVAE) and supervised approaches (DEAR) in terms of reconstruction quality. To our understanding, this superior performance is attributed to the specialized training strategy implemented in the framework.

Table 1: Quantitative comparison results with typical DRL approaches in FID and KID.

| Method | CelebA | | LSUN-horse | | LSUN-bedroom | |
|---|---|---|---|---|---|---|
| | FID $\downarrow$ | KID $\times 10^3 \downarrow$ | FID $\downarrow$ | KID $\times 10^3 \downarrow$ | FID $\downarrow$ | KID $\times 10^3 \downarrow$ |
| VAE [15] | 53.3 ± 0.6 | 51.4 ± 0.4 | 172.8 ± 1.7 | 181.7 ± 2.1 | 195.8 ± 4.1 | 226.4 ± 5.4 |
| $\beta$-VAE [15] | 136.2 ± 1.6 | 107.0 ± 2.7 | 272.4 ± 3.2 | 294.2 ± 5.3 | 288.1 ± 5.7 | 225.7 ± 6.0 |
| $\beta$-TCVAE [16] | 139.1 ± 0.8 | 113.2 ± 4.1 | 173.0 ± 4.8 | 217.35 ± 9.2 | 191.0 ± 5.0 | 179.2 ± 7.4 |
| FactorVAE [43] | 134.5 ± 0.3 | 92.0 ± 0.5 | 248.5 ± 5.5 | 155.3 ± 3.7 | 235.7 ± 3.2 | 172.8 ± 3.9 |
| DEAR [29] | 70.7 ± 0.3 | 52.6 ± 0.1 | 136.4 ± 1.6 | 113.7 ± 0.9 | 177.6 ± 3.5 | 157.8 ± 2.3 |
| **GEM (Ours)** | **46.0 ± 0.1** | **48.3 ± 0.2** | **101.0 ± 1.1** | **65.5 ± 1.7** | **125.4 ± 1.2** | **76.1 ± 1.1** |

As shown in Table 1, GEM surpasses baseline models in reconstruction quality on the datasets of CelebA, LSUN-horse, and LSUN-bedroom. However, the use of the disentanglement coefficient in the $\beta$-VAE branch leads to an inevitable trade-off in reconstruction quality, making the model less comparable to the models focused on generation quality (e.g., GAN and Diffusion [78]). Therefore, the integration with leading generative models can be a direction for our future work. For additional comparison results, please refer to Appendix A.1.

Table 2: Computational efficiency report in parameters size, FLOPs, memory cost and training time.

| Models | Params(M) | GFLOPs(B) | Mem(M) | TT(s) |
|---|---|---|---|---|
| FactorVAE | 55.9 | 3.8 | 200.5 | 63.9 |
| DEAR | 53.4 | 3.5 | 267.2 | 91.8 |
| GEM (Single) | 44.7 | 2.8 | 173.6 | 51.5 |
| **GEM (Full)** | 49.6 | 3.2 | 222.8 | 78.9 |

Furthermore, we evaluate four relation-aware models: FactorVAE, DEAR, GEM (Single), and GEM (Full), on quantitative comparisons of computational resources. Notably, GEM (Single) is the variant of GEM that incorporates single attribute determination branch (we only provide the initial relations to make it relation-aware). Table 2 shows that GEM outperforms DEAR and is comparable to FactorVAE on computational efficiency. This is mainly attributed to FactorVAE's utilization of a simple convolutional encoder, whereas GEM employs a $\beta$-VAE based encoder to strengthen disentanglement. In addition, the efficiency of full GEM is slightly inferior to GEM (Single), due to the extra modules for relation discovery and refinement.

## 4.3 Evaluations of Interpretability and Generalizability

As a by-product, GEM inherits the interpretability and generalizability of MLLMs. Theoretically, owing to the commonsense reasoning faculties of MLLMs, our model can be generalized to discover any attributes and interrelations across various real-world objects and scenes. To demonstrate the robustness and generalizability of GEM, we perform extra experiments on more complex scenes in LSUN, specifically targeting the typical object subset LSUN-horse and the scene subset LSUN-bedroom. Furthermore, we test the attributes beyond the 40 specified in CelebA, collectively showcasing the model's superiority. To highlight the characteristics of bidirectional weighted DisGragh, we intentionally select paired fine-grained attributes exhibiting inconsistent bidirectional relations.

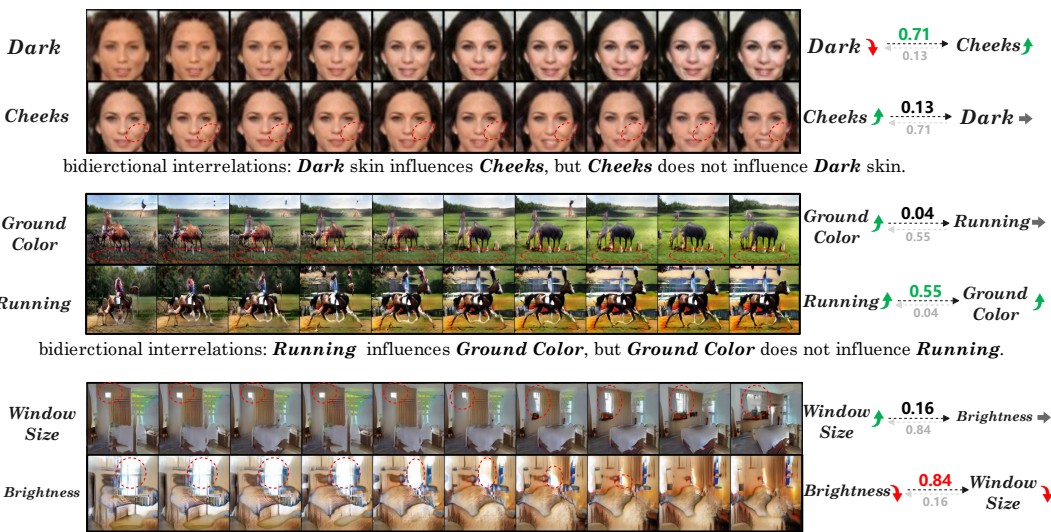

Figure 7: Relation-aware disentanglement results on LSUN and the attributes beyond CelebA. Paired fine-grained attributes with inconsistent bidirectional relations are chosen to indicate effectiveness.

As depicted in Figure 7, GEM successfully achieve fine-grained disentanglement on complex scenes, while identifying bidirectional and weighted relations among attributes. Furthermore, the artifacts observed in the results of LSUN datasets are mainly due to the datasets' clutter (evidenced by the increase of FID and KID scores in Table 1). Nonetheless, despite the ambiguous and challenging nature of the data, GEM still obtain commendable disentanglement outcomes, affirming its robustness.

## 4.4 Evaluations of MLLMs

Our model leverages the commonsense knowledge embedded in MLLMs to predict interrelations. This is predicated on the assumption that MLLMs, including their future iterations, are powerful and reliable enough to comprehend the physical rules of the real world (e.g., aging brings wrinkles, sunrise brings light, etc.). Therefore, before utilizing the interrelation discovery branch, it is imperative to evaluate the reliability of MLLMs. This evaluation guarantees that the identified interrelations and their associated impact scores are dependable and can be effectively applied to downstream modules. To this purpose, we evaluate three latest MLLMs including GPT-4o, GPT-4v and GLM-4—against the ground truth attributes of the CelebA dataset. The horizontal axis presents the targeted attributes selected from the CelebA, where the vertical axis presents the percentage of scoring accuracy.

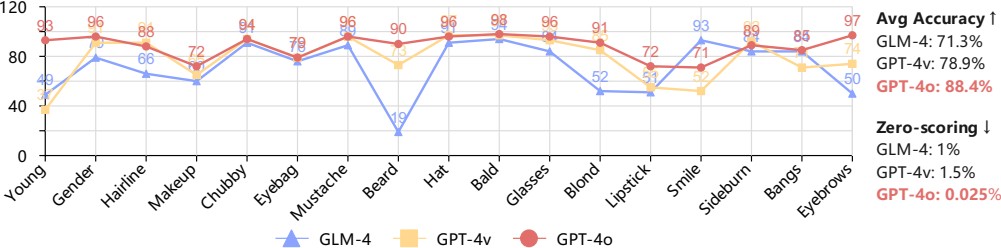

Figure 8: Evaluation experiments on the various MLLMs for attributes scoring.

As reported in Figure 8, GPT-4o outperforms other models on individual attribute scoring, achieving accuracy exceeding 90% for the majority of attributes. Specifically, it exhibits superior performance on attributes like *Beard*, *Young*, and *Eyebrows*, where other models yield significantly lower scores. In addition, GPT-4o achieves the highest average accuracy of 88.4% and the lowest zero-scoring rate at 0.25%, indicating a minimal rate of the meaningless predictions where all attributes are scored as zero. We conducted further evaluations on individual attributes, where GPT-4o also demonstrated superior performance (see Appendix A.7). Based on the evaluations, we employ GPT-4o as the interrelation predictor in the model.

## 4.5 Ablation Study

To analyze the effectiveness of individual components in GEM, we perform an ablation study focusing on the importance of the $\beta$-VAE based branch, GNN-based graph learner $E_{gnn}$, and adversarial training strategy. The CelebA dataset served as the experimental platform for the investigations. It is worth noting that the complete removal of $\beta$-VAE branch is infeasible, as it would prevent the model from extracting attributes. Therefore to evaluate the importance of independent attribute extraction, we replace the $\beta$-VAE with the vanilla VAE, which does not enforce the independence of factors.

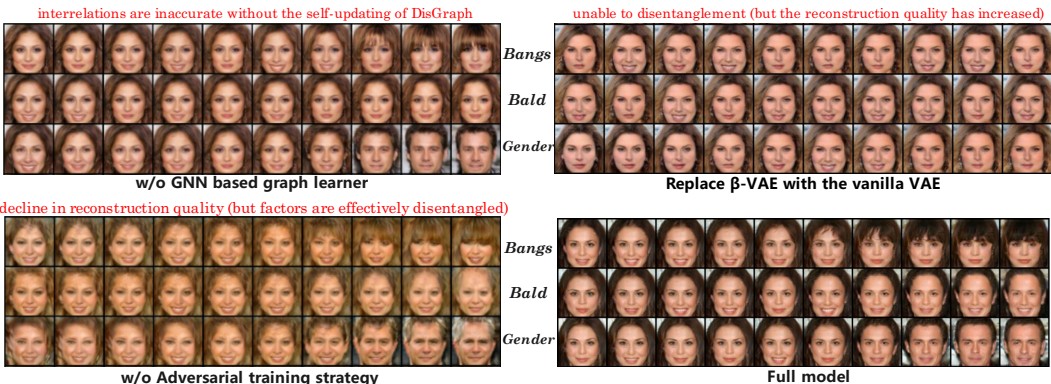

Figure 9: Ablation on replacing $\beta$-VAE with VAE, w/o graph learner, and w/o adversarial strategy.

As depicted in Figure 9, replacing $\beta$-VAE encoder results in a declined disentanglement capability, albeit with an improvement in reconstruction quality. In addition, the removal of GNN-based graph learner prevents the parameter updating of DisGraph, leading to the inaccurate determination of relations (e.g., the relation between *Bald* and *Gender* weakens). It is worth noting that the removal of both graph learner and initialization process within the framework precludes the learning of interrelations. Furthermore, eliminating the adversarial training strategy in GEM and relying solely on the standard VAE loss function results in a significant decline in reconstruction quality. The aforementioned results highlight the effectiveness of each part of our framework.

## 5 Conclusion

In this paper, we aim to explore the logical interrelations between semantic attributes within complex data, which is a critical challenge that existing DRL have yet to properly address. To this end, we introduce GEM, a $\beta$-VAE and MLLMs-based framework, designed to achieve fine-grained and relation-aware disentanglement. In this framework, DRL and MLLMs are integrated via a bidirectional and self-driven graph. Both qualitative and quantitative experiments demonstrate GEM's superior disentanglement and reconstruction capacities over typical DRL models. In addition, the model shows its enhanced interpretability and generalizability inherited from MLLMs.

## 6 Acknowledgments

This research is supported by the IDT Foundation of Youth Doctoral Innovation [Grant S203.2.01.32.002], the National Natural Science Foundation of China [Grant 62302246] and the Zhejiang Provincial Natural Science Foundation of China [Grant LQ23F010008].

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

# A   Appendix

## A.1   Discussion

**Disentanglement Ability.** As shown in Table 3, we evaluate the disentanglement ability of our model (DRL branch only) against typical DRL models. Our model outperforms them across numerous metrics. However, it is really meaningful and interesting to discuss "what is a better disentanglement".

Table 3: Disentanglement metrics among VAE, $\beta$-VAE and GEM.

| Models | MIG | IRS | MI | Info |
|---|---|---|---|---|
| VAE | 0.19 | 0.35 | 0.84 | 0.81 |
| $\beta$-VAE | 0.49 | 0.55 | 0.88 | 0.82 |
| **GEM** | **0.53** | **0.60** | **0.86** | **0.85** |

If it means the better performance on independently decomposing factors, then the inclusion of interrelations might not seem beneficial; however, if it refers to a better performance/practicality for real and complex scenarios, our disentanglement paradigm excels by statistically capturing the logical rules of real world. Specifically, the inclusion of interrelations can be beneficial in model generalizability, counterfactual reasoning and practical usages.

**Trade-off between Quality and Interpretability.** Even though our model achieved superior performance among DRL approaches, an inevitable trade-off between reconstruction and disentanglement remains, resulting in decreased reconstruction quality compared to the models focused on generation quality such as GANs and Diffusions (see Table 4).

Table 4: Quantitative comparison results with leading image generation models in FID and KID.

| Method | CelebA ($64\times64$) | | CelebA ($256\times256$) | |
|---|---|---|---|---|
| | FID $\downarrow$ | KID $\times10^3 \downarrow$ | FID $\downarrow$ | KID $\times10^3 \downarrow$ |
| **GEM (Ours)** | 46.05 | 48.32 | 50.93 | 51.01 |
| Vanilla VAE | 53.39 | 51.48 | 56.82 | 61.26 |
| StyleGAN2 (40k steps) | 12.94 | 9.20 | 18.02 | 19.55 |
| DDPM (Diffusion, $T = 1$k) | 8.56 | **6.56** | 15.93 | 10.01 |
| DDIM (Implicit Diffusion, $T = 1$k) | 10.04 | 8.15 | 16.24 | 13.62 |
| Stable Diffusion (fine-tuning) | **7.72** | 7.22 | **10.63** | **9.17** |

Since our model is oriented towards interpretability, we consider this trade-off acceptable. However, it is insightful to leverage the advantages of both DRL and non-DRL models within a mutually beneficial closed-loop architecture, and we will make efforts to improve our work in this direction.

**Current Limitations.** Compared to existing DRL approaches, GEM emphasizes discovering underlying interrelations between attributes. This logical and effective framework can benefit a wide range of downstream tasks and practical applications such as controllable generation, medical image analysis, and automatic driving. However, there are still some limitations to this method. Firstly, the trade-off between reconstruction quality and disentanglement capacity, as a common challenge in the domain, is still not properly addressed in this work. To tackle it, we are currently investigating the integration of powerful generative models, e.g., diffusion models [78] and visual auto-regressive models [79], into our framework. Secondly, the current implementation of GEM is not designed to work with 3D data, where 3D representations are much more complex. To understand our real world, it would be necessary to enhance the model with specific improvements to handle 3D scenes.

## A.2   Dataset details

**CelebA.** The Celebrity Faces Attributes (CelebA) dataset [75] is a widely-used large-scale face attributes dataset that contains more than 200,000 celebrity images, each annotated with 40 attributes.

These annotations cover a wide range of facial attributes such as 'smiling', 'wearing Hat', 'young', 'wavy Hair', 'male', and 'mustache'. These attributes are labeled as present or absent in each image. This dataset is designed for various DRL and computer vision tasks, such as face recognition, face attribute disentanglement, and face editing. Commonly, we use employ the entirety of the CelebA dataset, which includes 162,770 images for training, 19,867 for validation, and 19,962 for testing.

**LSUN.** The Large-scale Scene Understanding (LSUN) dataset [76] is a comprehensive collection for deep learning and computer vision, widely used in the domain of perceptual analysis and attribute disentanglement. This dataset consists of around one million labeled images for each of 10 scene categories and 20 object categories. The scene categories include diverse environments such as bedrooms, conference rooms, dining rooms, kitchens, living rooms, and etc. The object categories in LSUN include horse, car, church, etc. We select a typical subset, LSUN-horse, to demonstrate the model's generalizability. We believe these two datasets are diverse enough to verify our GEM.

### A.3 Baseline details

#### A.3.1 $\beta$-VAE

The $\beta$-VAE [15] is an extension of the standard VAE [14], introducing an adjustable hyperparameter $\beta$ prior to the KL term in vanilla VAE:

$$\mathcal{L}(\theta, \phi) = \mathbb{E}_{q_\phi(\mathbf{z}|\mathbf{x})} \left[ \log p_\theta(\mathbf{x}|\mathbf{z}) \right] - \beta D_{KL} \left( q_\phi(\mathbf{z}|\mathbf{x}) \| p_\theta(\mathbf{z}) \right) \tag{12}$$

where the penalty coefficient $\beta$ balances the disentanglement capacity and reconstruction quality. When $\beta > 1$, this penalty increases the emphasis on learning disentangled representations in the latent space. However, increasing $\beta$ can cause a trade-off on the reconstruction quality.

#### A.3.2 $\beta$-TCVAE

The Total Correlation beta-VAE ($\beta$-TCVAE) [16] builds upon the $\beta$-VAE to further enhance the disentanglement of latent representations. It achieves this by the reduction of the total correlation (TC) term, extracted from the KL divergence term:

$$\mathbb{E}_{p(x)}[KL(q(z \mid x)\|p(z))] = KL(q(z,x)\|q(z)p(x)) + \beta KL \left( q(z)\| \prod_j q(z_j) \right) + \sum_j KL \left( q(z_j) \| p(z_j) \right) \tag{13}$$

where $j$ represents the dimension of latent code $z$. The penalty coefficient $\beta$ is selectively applied to the second term, i.e. TC term, on the right side of the loss function. It aims to make latent variables statistically independent of each other, therefore enhancing disentanglement. We include $\beta$-TCVAE in the quantitative comparisons, following the official implementation.

#### A.3.3 FactorVAE

The FactorVAE [43] is a typical variant of VAE, which employs a discriminator network in an adversarial manner to accurately estimate and minimize the total correlation term in the loss function. This adversarial strategy further enforces the factorization of the latent space, leading to improved disentanglement. We include $\beta$-VAE, $\beta$-TCVAE, FactorVAE in the quantitative comparisons following their official implementation. We evaluate the reconstruction quality by employing the Fréchet Inception Distance (FID) and Kernel Inception Distance (KID) metrics for comparative assessment.

#### A.3.4 DEAR

DEAR [29] is the weakly supervised structural disengagement framework. It facilitates causal representation learning by adopting a structural causal model (SCM) as the prior distribution. This SCM prior is supervised by the information on the ground-truth factors and their underlying causal structure from the database. We integrate DEAR into the quantitative and qualitative comparisons, training it with the annotations provided by the CelebA. All in all, we evaluate our unsupervised framework against both unsupervised ($\beta$-VAE, $\beta$-TCVAE, FactorVAE) and supervised (DEAR) DRL methods, thereby rigorously evaluating the capabilities in reconstruction and disentanglement.

### A.4 Face landmark results

Landmark detection are algorithm and technique utilized to detect and track specific key points in the image or video, encompassing a wide range of applications across various fields such as

computer vision, robotics, and geospatial analysis. In this work, we employ landmark detection as a pre-processing method to extract the key points of the main object, thus removing the redundant parts in the image through cropping and resizing.

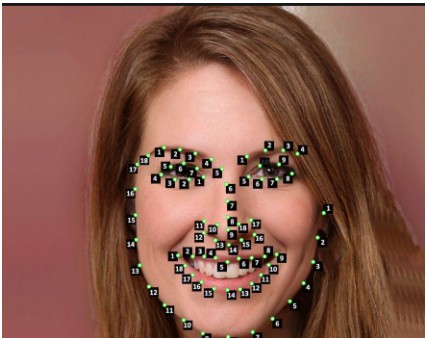

Figure 10: 68-points landmark pre-processing for data from the CelebA.

We introduce the pre-processing phase for CelebA, where 68 landmark points are identified and extracted commonly. As shown in Figure 10, points 1 to 17 present the jawline, points 18 to 27 for the eyebrows, points 28 to 36 for the nose, points 37-48 for the eyes, and points 49-68 identify the lips.

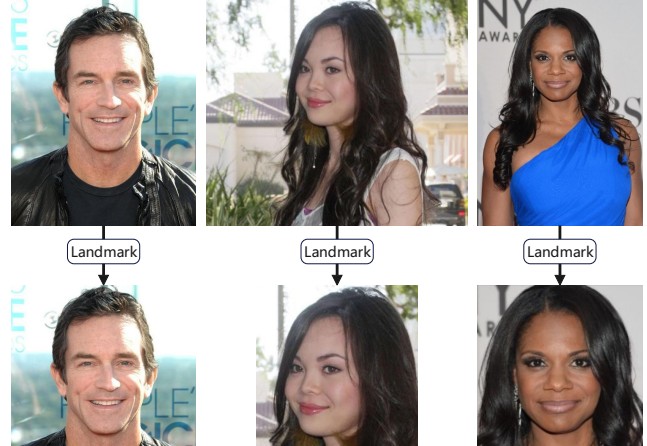

Figure 11: Image normalization based on the landmark detection.

Figure 11 illustrates some results of the pre-processing block. The normalized images benefit subsequent factor disentanglement and interrelation discovery.

## A.5 Details of adversarial training strategy

As described in Section 3.1, the prior assumption that $p_\theta(z) \sim \mathcal{N}(0, 1)$ no longer holds, we reformulate the $\beta$-VAE loss as:

$$
\begin{aligned}
\mathcal{L}(\theta, \phi) &= \mathbb{E}_{x \sim q_\phi(x)} (\mathbb{E}_{z \sim q_\phi(z|x)} \left[\log p_\theta(x|z)\right] - \beta D_{KL}\left(q_\phi(z|x) \| p_\theta(z)\right)) \\
&= \mathbb{E}_{x,z \sim q_\phi(z,x)} \left[\log p_\theta(x|z)\right] - \beta \mathbb{E}_{x,z \sim q_\phi(z,x)} log\left(\frac{q_\phi(z|x)}{p_\theta(z)}\right)
\end{aligned}
\tag{14}
$$

Subsequently, the gradient of $\mathcal{L}$ with respect to $\theta$ can be derived as:

$$\nabla_\theta \mathcal{L}(\theta, \phi) \stackrel{x=D_\theta(z)}{=} \mathbb{E}_{z\sim q(z)}[\nabla_x log(p_\theta(x|z))\nabla_\theta x - \beta\nabla_x log\left(q_\phi(z|x)/p_\theta(z)\right)\nabla_\theta x]$$

$$= \mathbb{E}_{z\sim q(z)}\nabla_x[log(p_\theta(x|z)) - \beta log(\frac{q_\phi(z|x)}{p_\theta(z)}]\nabla_\theta x$$

$$= \mathbb{E}_{z\sim q(z)}\nabla_x[log(p_\theta(x,z)) - \beta log(\frac{q_\phi(z,x)}{q_\phi(x)})]\nabla_\theta x$$

$$= \mathbb{E}_{z\sim q(z)}\nabla_x[log(p_\theta(x,z)) - \beta log(\frac{q_\phi(z,x)}{q_\phi(x)})]\nabla_\theta x \tag{15}$$

$$= \mathbb{E}_{z\sim q(z)}\nabla_x[log(\frac{p_\theta(x,z)}{\beta q_\phi(x,z)})]\nabla_\theta x + \beta\mathbb{E}_{z\sim q(z)}\nabla_x q_\phi(x)\nabla_\theta x$$

where the first term on the right side needs to be approximated by a neural network due to the altered assumption. Therefore, we define:

$$p(x,z) = p((x,z)\in E_\phi)p(x,z|(x,z)\in E_\phi) + p((x,z)\in D_\theta)p(x,z|(x,z)\in D_\theta)$$
$$= p((x,z)\in E_\phi)q_\phi(x,z) + p((x,z)\in D_\theta)p_\theta(x,z) \tag{16}$$

then

$$p((x,z)\in E_\phi|x,z) = \frac{p((x,z)\in E_\phi)p(x,z|(x,z)\in E_\phi)}{p(x,z)}$$

$$= \frac{1}{1+\alpha\frac{q_\phi(x,z)}{p_\theta(x,z)}}$$

$$p((x,z)\in D_\theta|x,z) = \frac{p((x,z)\in D_\theta)p(x,z|(x,z)\in D_\theta)}{p(x,z)} \tag{17}$$

$$= \frac{1}{1+\frac{1}{\alpha}\frac{p_\theta(x,z)}{q_\phi(x,z)}}$$

$$\alpha = \frac{p((x,z)\in D_\theta)}{p((x,z)\in E_\phi)}$$

Given the controllable proportion of the input images, denoted by $\alpha \in [0,1]$, we define:

$$D(x,y) = log(\frac{p_\theta(x,z)}{\beta q_\phi(x,z)}) \tag{18}$$

then

$$\frac{p_\theta(x,z)}{\beta q_\phi(x,z)} = e^{D(x,z)} \tag{19}$$

where $\beta \in (1, +\infty)$ is proposed to enhance the disentanglement ability of $\beta$-VAE. We subsequently define the variable $k$:

$$suppose\ \ k = \frac{\alpha}{\beta}$$

$$p((x,z)\in E_\phi|x,z) = \frac{1}{1+ke^{-D(x,z)}}$$

$$p((x,z)\in D_\theta|x,z) = \frac{1}{1+\frac{1}{k}e^{D(x,z)}} \tag{20}$$

$$since:\ \ 1 = p((x,z)\in E_\phi|x,z) + p((x,z)\in D_\theta|x,z)$$

Given that the solution for $k$ is determined to be 1, under the condition that $\alpha = \beta = 1$ (otherwise $\forall(x,z)\ D(x,z) = 0$ which is obviously impossible), it is logically imperative to require distribution $p(x,z|(x,z)\in E_\phi)$ and $p(x,y|(x,z)\in D_\theta)$ to closely approximate $p(x,z)$. On this basis, we found:

$$p(x,z) = p(x,z|(x,z) \in E_\phi) \iff p((x,z) \in E_\phi|x,z) = p((x,z) \in E_\phi)$$
$$p(x,z) = p(x,z|(x,z) \in D_\theta) \iff p((x,z) \in D_\theta|x,z) = p((x,z) \in D_\theta) \tag{21}$$

where if $\alpha = 1$, we attain the optimization objective of $p((x,y) \in D_\theta) = p((x,y) \in E_\phi) = \frac{1}{N_m}$. To this end, we can finally use the cross-entropy algorithm to optimize $D$:

$$
\begin{aligned}
\mathcal{L}_{adv} = \mathcal{L}(D) &= cross\_entropy(p_{gt}, p_{\theta,\phi}) \\
&= -\frac{1}{N_{bc}} \sum_{i=0}^{N_{bc}} [p((x_i, z_i) \in E_\phi)logp((x_i, z_i) \in E_\phi|x,z) + p((x_i, z_i) \in D_\theta)logp((x_i, z_i) \in D_\theta|x,z)] \\
&= -\frac{1}{N_{bc}N_m} \sum_{i=0}^{N_{bc}} [-softplus(-D(x_i, z_i)) - softplus(D(x_i, z_i))] \\
&= \frac{1}{N_{bc}N_m} \sum_{i=0}^{N_{bc}} [softplus(-D(x_i, z_i)) + softplus(D(x_i, z_i))] \\
&= \frac{1}{N_{bc}N_m} [\sum_{i=0;(x_i,z_i) \in E_\phi}^{N_{bc}} softplus(-D(x_i, z_i)) + \sum_{i=0;(x_i,z_i) \in D_\theta}^{N_{bc}} softplus(D(x_i, z_i))],
\end{aligned}
\tag{22}
$$

where $softplus(x) = \ln(1 + e^x)$ is a smooth activation function. $N_{bc}$ and $N_m$ represent the number of samples and posterior samples in a batch. The additional adversarial loss ensures the maintenance of reconstruction quality, tending to diminish as the capacity for disentanglement increases.

## A.6 Details of interrelation determining strategy

In the interrelation discovery branch, we propose the Somers' Delta (Somers' D) [71] algorithm to determine and rank the bidirectional relations among the attributes extracted by the $\beta$-VAE branch. Somers' D is a statistical measure used to assess the strength and direction of the relation between variables. It is a nonparametric measure that can be considered a measure of rank correlation, similar to Kendall's tau [72], but with a focus on asymmetric relations.

Specifically, the calculation of impact scores based on Somers' D is upon the number of concordant pairs ($C$) and discordant pairs ($D$). The formula of Somers' D on variable $Y$ and $X$ can be given as:

$$D_{YX} = \frac{C - D}{C + D + T_x}, \quad D_{XY} = \frac{C - D}{C + D + T_y} \tag{23}$$

where $T_x$ is the number of ties only for the independent variable $X$, and $T_y$ is the number of ties only for the independent variable $Y$. The obtained attribute scores and interrelations from MLLMs are evaluated. For example, suppose we have the sample dataset $S = (1,2), (3,1), (2,3)$:

| Variable | Pairs | Value |
|----------|-------|-------|
| $N_c$ | (1,2) vs (2,3) | 1 |
| $N_d$ | (1,2) vs (3,1) and (2,3) vs (3,1) | 2 |
| $N_y$ | None | 0 |

Then the Somers' D indicator can be calculated as follows:

$$D = \frac{N_c - N_d}{N_c + N_d + T_y} = \frac{1 - 2}{1 + 2 + 0} = -\frac{1}{3} \tag{24}$$

This obtains a value of approximately -0.33, signifying a negative correlation between variables $X$ and $Y$. This calculation demonstrates that the Somers' D metric is straightforward to calculate and

is particularly applicable to ordinal variables. Furthermore, Somers' D is asymmetric and capable of distinguishing bidirectional relationships between variables. These characteristics make it highly suitable for integration into our model.

## A.7 Additional results

We perform additional evaluation results for individual attributes on different MLLMs with Ground Truth (GT) labels in CelebA. As shown in Figure 12, the results demonstrate the reliability of the GPT-4o employed in our work.

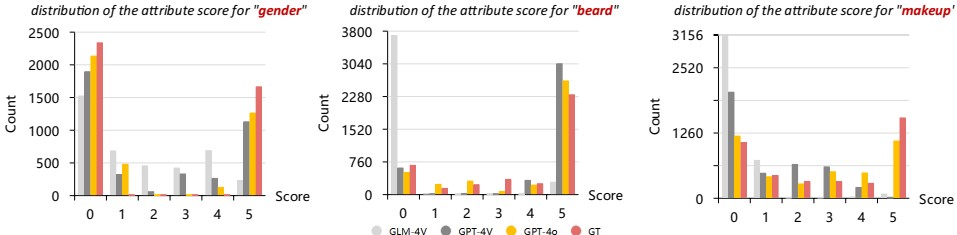

Figure 12: Reconstruction results of the CelebA.

We also present additional results that demonstrate the capacity of GEM in fine-grained and relation-aware disentanglement. Figure 13, Figure 14, Figure 15 demonstrates the reconstruction results on the CelebA, LSUN-bedroom and LSUN-horse, respectively. It is obvious that following the landmark pre-processing, GEM effectively identify the main part of facial images while mitigating noise. This enhancement facilitates the downstream processes of the model. In addition, as depicted in Figure 16, GEM is capable of processing high-definition images given sufficient computational resources.

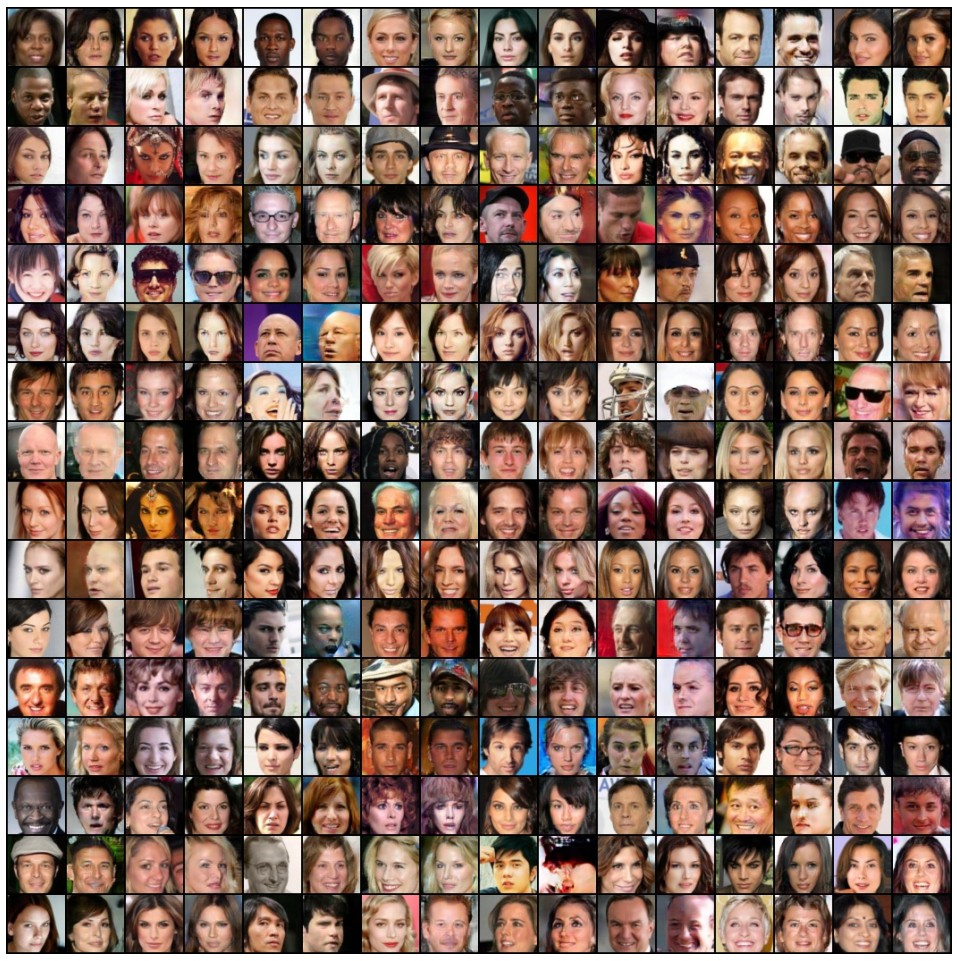

Figure 13: Reconstruction results of the CelebA.

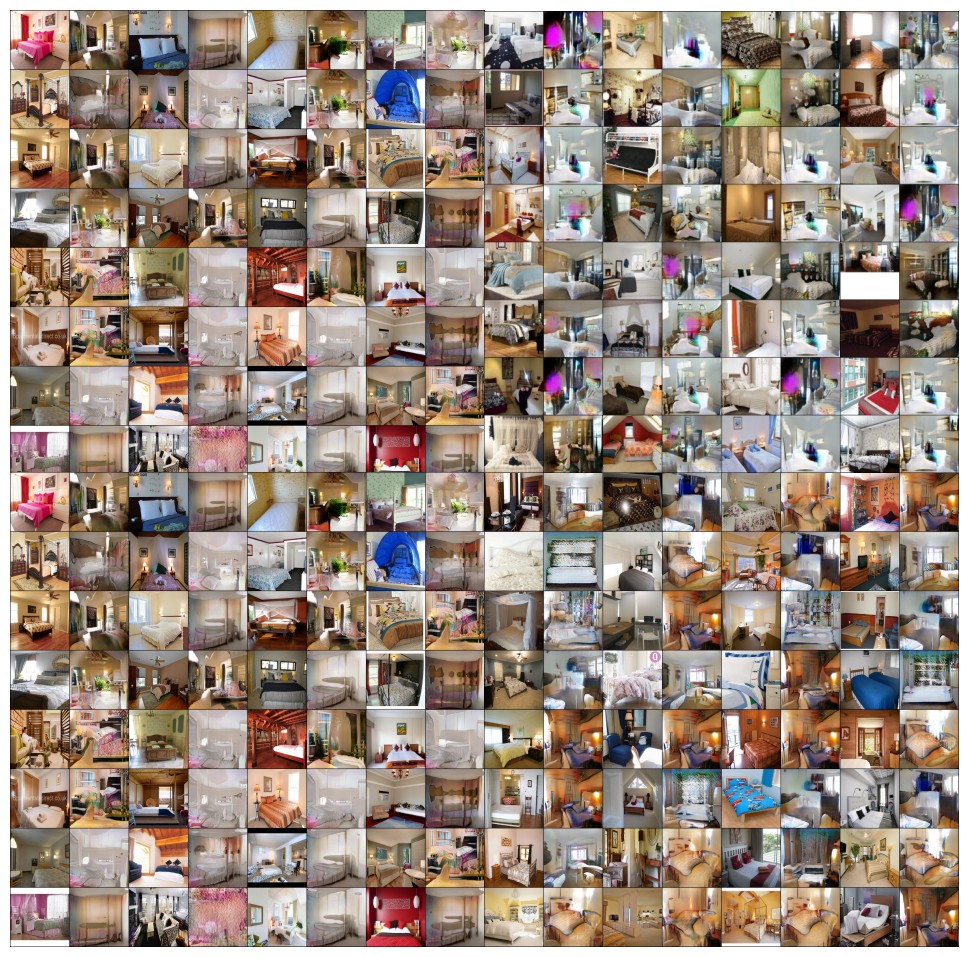

Figure 14: Reconstruction results of the LSUN-bedroom.

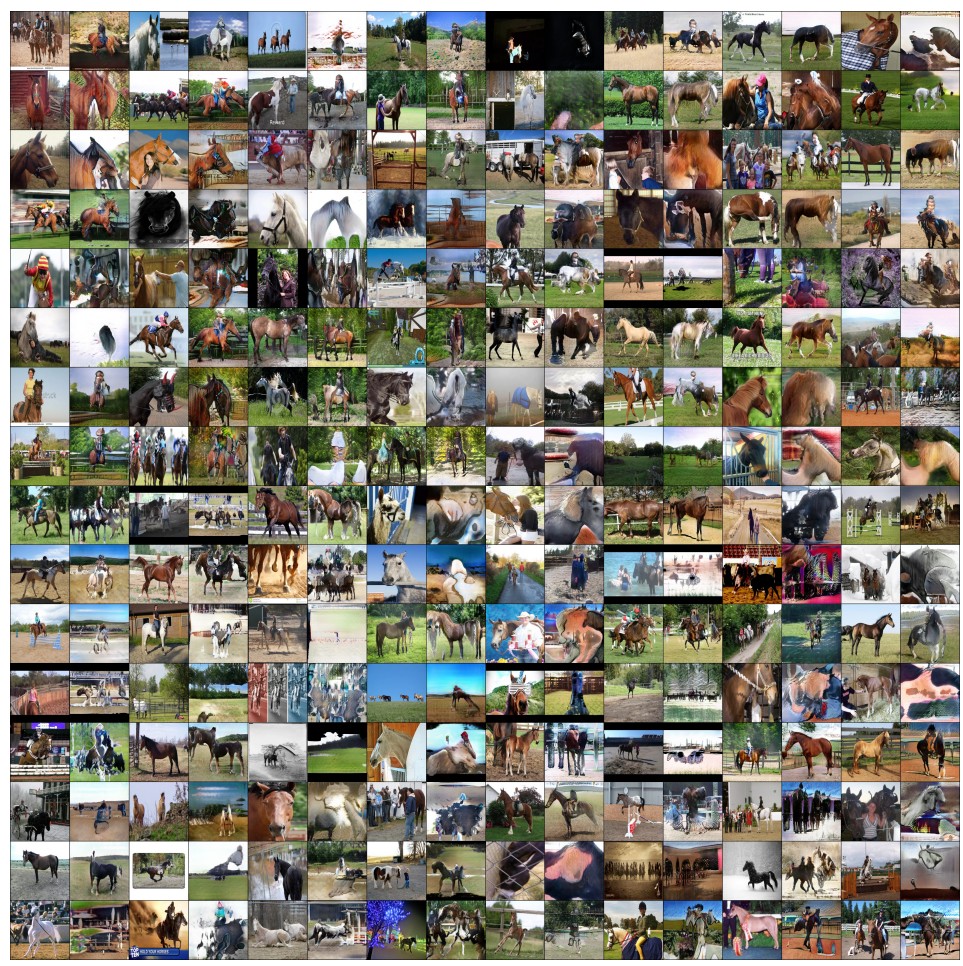

Figure 15: Reconstruction results of the LSUN-horse.

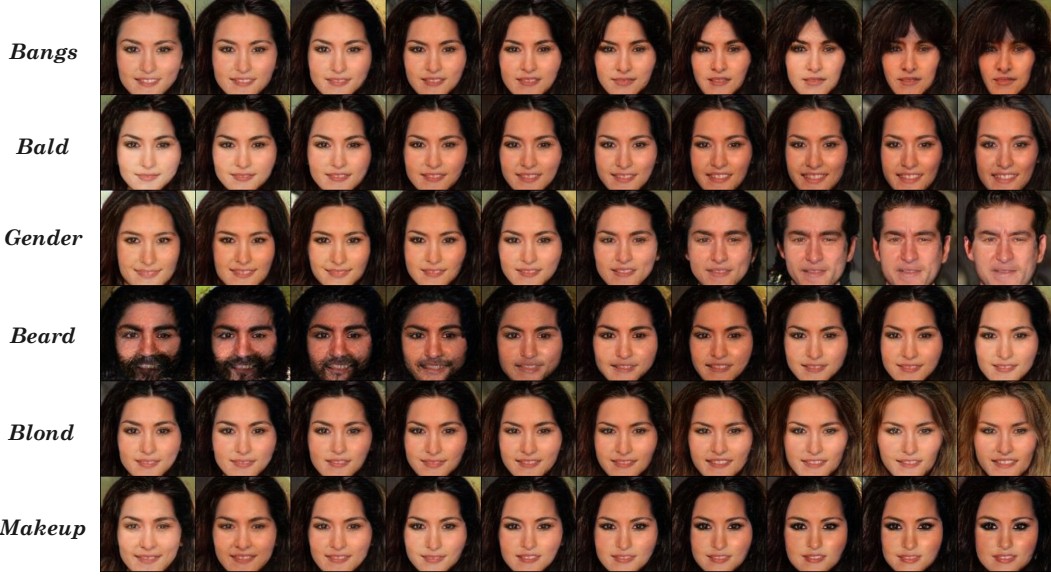

Figure 16: high-definition results at 256×256 on the CebebA.

