# OpenReview forum: "Graph-based Unsupervised Disentangled Representation Learning via Multimodal Large Language Models"
_NeurIPS.cc/2024/Conference — NeurIPS 2024 poster_

### Official Review · Reviewer_Y3bu · 2024-06-29

**Soundness:** 2
**Presentation:** 2
**Contribution:** 2
**Rating:** 6
**Confidence:** 4

**Summary:**

This paper introduces a bidirectional weighted graph-based framework, to learn factorized attributes and their interrelations within complex data. The authors proposed a β-VAE based module to extract factors as the initial nodes of the graph, and leverage the multimodal large language model (MLLM) to discover and rank latent correlations, thereby updating the weighted edges. Experiments demonstrate evidence of effective performance in disentanglement and reconstruction.

**Strengths:**

This paper integrates the VAE with multimodal large language model, which provides strong interpretability via the open-world knowledge of the large language models/multimodal models.

**Weaknesses:**

* As a generative model, β-VAE is extensively discussed with many useful variants. While this work heavily borrows the existing advances of  β-VAE, I would respect ground-breaking findings or improvements on this model (e.g., closed-form derivation of the feature distribution z after employing a graphical model). Otherwise speaking about absolute performance, I expect a standard diffusion model can easily outperform β-VAE and most of its variants (And authors should include comparison and discussion with diffusion models, even it is less likely that the proposed method can outperform, but insights on what β-VAE is lacking would be insightful, although it’s less meaningful in the era of diffusion model).
* Related to above, the technical novelty and motivation of this work is limited since the current model can be easily understood as the combination of β-VAE and Large Multimodal models. There is less strong justification while this combination is optimal rather than other obvious choices (e.g., integrating a vision-language model, or diffusion+MLLM (since diffusion models are also trained with variational inference)). Author should either provide strong justification (better if theoretical) or more empirical validation showing the proposed framework is optimal than other obvious variants to enhance their motivation and novelty.
* Line 148, it is obvious that z dos not follow multivariate normal, and author proposed Eq. 4 as the solution. Surprisingly, the author only showed the derivation of this equation (which is trivial in my view) in the appendix, with no justification why this is optimal. Why α=1? Or how strong is this assumption? Is the resulting objective still convex? What is the distribution of z under the new objective? Overall the authors imposed made many conditions in the ease of derivation with many justifications missing.
* The authors only used GCN to learn the graph representation, which ruled out many other prominent choices (E.G., GAT, gatV2, GIN, GTN, GraphSAGE). authors should conduct thorough ablation study to select the optimal GNN architecture.
* As an applied paper, it is less acceptable that codes are not released in the current phase. As many papers release their codes in anonymous GitHub repos and recently there are higher requirements of reproducibility and codeavilability in top venues. We also need to check the overall soundness of the empirical implementation.

**Questions:**

* For the bidirectional weighted graph, it is unclear why this is a good choice. From my knowledge, a causal DAG would be a better choice to represent the interrelation of the entities, and there are many works (e.g., [1]) available deriving the closed-form distributions of the features under this assumption.

[1] BayesDAG: Gradient-Based Posterior Inference for Causal Discovery. NIPS 2023.

**Limitations:**

The authors addressed some limitations of their works.

---

> ### Author Rebuttal · Authors · 2024-08-07
>
> *We greatly appreciate your insightful comments and commit to refining our manuscript based on your suggestions. Below, we address all your concerns.*
>
> **W1: Why not replace $\beta$-VAE with advanced generative models**
>
> **R1:** Thanks for your insightful comments to help us improve our work. Firstly, we would like to discuss our standpoint on using other generative models, especially the Diffusion you mentioned. In our framework, the core of $\beta$-VAE based encoder $E_{sem}$ is to extract and initialize disentangled latent factors. **This important perception block cannot be ideally replaced by Diffusion or other non-DRL generative models, as they do not produce an orthogonal and disentangled latent space**, and thus can not comprehend and extract disentangled factors (even though some of them are also trained with variational inference).
>
> Contrary to replacing $\beta$-VAE with the Diffusion (or other variants) as the semantic encoder, we tend to integrate it as the decoder $D_{rec}$.  This integration capitalizes on the superior generative abilities of the Diffusion model to improve the reconstruction fidelity of our framework. Concomitantly, the enhanced perceptual capacity of our model can, in turn, refine the generative process of the Diffusion, **thereby establishing a synergistic, mutually beneficial closed-loop**. We are currently implementing this feature. However, should time constraints arise, we promise to at least include an extra discussion section to cover these content.
>
> **W2: Lacking technical novelty and the justifications of technical combinations**
>
> **R2**: First, **we must emphasize that the technical novelty and motivation of this work** is: 1) We are first to leverage the commonsense reasoning of MLLMs to discover and rank the semantic interrelations for DRL; 2) We propose a novel and practical disentanglement framework built upon β-VAE and MLLMs, with a bidirectional graph architecture,  specifically designed to learn the interrelations between independent factors.
>
> The naive combination of diffusion+MLLM could not achieve the same purpose of learning relation-aware representations, thereby facilitating practical and controllable disentanglement. This shortfall arises because Diffusion models are not designed for disentanglement, as they do not produce an orthogonal and disentangled latent space. Therefore, it is hard to directly compare our proposed method with the combinations you mentioned.
>
> **W3: Unclearness of equations and justifications in Section 3.1**
>
> **R3**: It's appreciated for your valuable comments. We apologize for not detailing the assumptions in the manuscript, and we promise to expand both the paper and the appendix to clarify these conditions and justifications explicitly.
>
> Specifically, regarding the question "Is the resulting objective still convex?", we assumed that the model is upper bounded by the norm of its gradient, which satisfies the Polyak-Lojasiewicz (PL) condition, thus ensuring the suboptimality of the model. The PL condition is weaker than (strong) convexity, meaning it can be applied in a broader range of scenarios. Existing models such as L1-regularized linear regression and logistic regression have been proven to satisfy this condition, which supports our decision to use logistic regression for gradient fitting.
>
> The assumption that $\alpha$=1 can be achieved by controlling the ratio of real samples to latent space samples. In the preliminary stage, we have tested model performance across a spectrum of hyper-parameters $\alpha$. Our empirical findings revealed a notable degradation in the quality of generated images when $\alpha$ was below 0.2 or above 5.5. Conversely, the KID metric exhibited stable consistency at $\alpha$=1 for most scenarios.
>
> And for your last concern: Given our utilization of the discriminator for gradient fitting, the distribution of $z$ under the new objective does not affect the final optimization procedure. Consequently, we did not delve into the new distribution properties of $z$.
>
> **W4: Authors should conduct thorough ablation study to select the optimal GNN architecture**
>
> **R4:** Thanks for your suggestions. We will consider potentially prominent graph choices with experimental proof, provided they can be optimized within our framework in an unsupervised manner.
>
> **W5: Authors should release their code**
>
> **R5**: Sure, we have released our project and sent the anonymous GitHub link to the AC as required. We highly welcome fellow researchers to follow and help improve our work.
>
> **Q1: For the bidirectional weighted graph, it is unclear why this is a good choice.**
>
> **R6**: Thank you for the feedback. As described in Section 2.2 and Figure 1, we analyzed the reasons for proposing DisGraph for embedding knowledge instead of using a causal graph: **1) the causal relationship is often overly simplistic, typically represented as binary and impractical.** However, in practice, paired variables commonly exhibit bidirectional influence, each impacting the other to varying degrees. For example, an increase in "age" can positively influence "baldness", whereas increased "baldness" does not significantly affect "age" in return;  2) Causal learning approaches are designed to model an event based on causal inference. **However, our setting, akin to most DRL approaches, is to model the scenario within the observed image. When considering the interrelations between observed vision attributes, the binary causal relation are inadequate.**  Furthermore, we conduct the comparison results with a causal model DEAR in Section 4.1, to show the experimental superiority of our choice. Regarding your comments, we recognize that the detailed explanations of our advantages over structured DRL (Hierarchical DRL, Causal DRL, etc.) were not sufficiently clarified. We commit to refine this aspect in our revision.

---

> > ### Comment · Reviewer_Y3bu · 2024-08-07
> >
> > I thank the authors for their response. However, I think my messages were poorly delivered and many of my intended points were lost when the authors summarize my comments. Sorry to say but the authors are just reiterating their main contributions and strengths of their methods instead of utilizing this chance to address the key concerns. Some cases to illustrate:
> >
> > 1. For W2 I was asking why this combination is optimal to **all** non-trivial combinations. The proper way should be listing some potential non-trivial combinations and show (better with experiments) why these combinations would underperform. This would greatly enhance the motivations of this work. Sadly, the authors just restate their contributions and novelty and choose only one example (diffusion+MLLM) to illustrate, which is less convincing.
> >
> > 2. For W1, why not replace the $\beta$-VAE with advanced generative models is just one of my questions, while my major point is why diffusion-based models are included as a baseline. The authors have 7 days and I think it shouldn't be difficult to perform an image-generation experiment with a standard diffusion model. The author should also address this question on absolute generation performance, and how much the proposed method is worse than (if any) the standard diffusion model to sacrifice for the interpretability. These empirical evaluations should be performed to more holistically evaluate the proposed method.
> >
> > I am open to challenges if I am wrong. But I cannot give a higher score based on the current version of the manuscript and authors' rebuttal. I think the authors should be candid when rephrasing the reviewers' comments, and make sure that every point raised is addressed properly.

---

> > > ### Author Response · Authors · 2024-08-11
> > >
> > > Thanks for your quick and patient feedback. We apologize for our misunderstandings regarding your comments and hope to address your concerns effectively in this renewed discussion.
> > >
> > > **1**: We would like to address your concerns of the optimal combination from both experimental and theoretical perspectives: 1) **Experimentally**, we have conducted extra experiments focusing on **relation-aware disentanglement** among typical DRL+MLLM/VLM combinations. The results are illustrated in the attached PDF accordingly:
> > >
> > > **Table 1**
> > >
> > > | Combinations| Results| Brief Comments|
> > > | --- | --- | --- |
> > > | $\beta$-VAE + GLM-4v | **Figure A3** (in attachment) | Producing counterfactual and anomalous results.|
> > > | FactorVAE + GPT-4o| **Figure A5** (in attachment)  | Suffering from inferior reconstruction quality.|
> > > | $\beta$-TC-VAE+ GPT-4o| **Figure A5** (in attachment)  | Failing to learn attributes effectively.|
> > > | $\beta$-VAE+ GPT-4o| **Figure A5** (in attachment)  | Current combination with most stable and effective performance|
> > > |The optimal selection of MLLM| **Figure A1** (in attachment) |GPT-4o outperforms others on attribute identification (also see Figure 6 in paper)|
> > >
> > > The visualization results demonstrate that the current combination yields the most stable and effective outcomes. We have also performed comparative experiments on **generation quality metrics** to evaluate the model performance (please refer to Table 2). 2) **Theoretically**, to our knowledge, these results can be attributed to the straightforward and effective design of the $\beta$-VAE. Since our model just leverages the DRL model to extract and initialize semantic factors, extensional designs can lead to instability (e.g., extra MLP classifier and discriminator in FactorVAE; the inflexible penalty in TC-$\beta$-VAE; the embedding codebook in VQ-VAE, etc.). These components, while potentially beneficial in certain contexts, may introduce unnecessary complexity and reduce the stability in our setting for relation-aware disentanglement.
> > >
> > > *The exclusion of non-DRL combinations (Diffusion, GAN, etc. + MLLMs) from **disentanglement capability comparisons**, is due to **their inability to generate orthogonal and disentangled latent spaces,** as detailed in **R1**. However, the comparisons of generation quality are conducted involving DRL and non-DRL models, for evaluating the model trade-off between generation and interpretability (see next response).*
> > >
> > > **2**. According to your response, we have included Diffusion and GAN models as the baselines in the generation quality experiments as shown in Table 2.
> > >
> > > **Table 2**
> > >
> > > | Model| CelebA (64x64)|  | CelebA (256x256) | |
> > > | --- | --- | --- | --- | --- |
> > > | | FID $\downarrow$| KID x $10^3$ $\downarrow$ | FID $\downarrow$| KID x $10^3$ $\downarrow$ |
> > > | **FactorVAE + GPT-4o**| 112.08 | 101.54 | 126.58 | 130.12 |
> > > | **$\beta$-TC-VAE + GPT-4o** | 68.17 | 62.90 | 91.45 | 87.22 |
> > > | **GEM (Ours)** | 46.05 | 48.32 | 50.93 | 51.01 |
> > > | **Vanilla VAE** | 53.39 | 51.48 |56.82 | 61.26 |
> > > | **StyleGAN2** *(40k steps)*| 12.94| 9.20| 18.02 | 19.55  |
> > > | **DDPM** *(Diffusion, $T$ = 1k)*| 8.56| **6.56** | 15.93 | 10.01 |
> > > |**DDIM** *(Implicit Diffusion, $T$ = 1k)*| 10.04 | 8.15 | 16.24 |  13.62 |
> > > | **Stable Diffusion** *(fine-tuning)*| **7.72** | 7.22 | **10.63** | **9.17** |
> > >
> > > *Due to time constraints, we present the results on the CelebA dataset, and we promise to provide comprehensive evaluations in the manuscript.*
> > >
> > > Even though our model achieved superior performance among DRL approaches, an inevitable trade-off between reconstruction and disentanglement remains, resulting in decreased reconstruction quality compared to image generation models (GAN, Diffusion, etc.). Since our model is oriented towards interpretability, we consider this trade-off acceptable (see Lines 253-257 in the paper). However, it is insightful to leverage the advantages of both DRL and non-DRL models within a mutually beneficial closed-loop architecture (as detailed in **R1**), and we will make efforts to improve our work in this direction.
> > >
> > > We are open to further discussions if you have any unresolved concerns. And we will conduct a comprehensive evaluation and analysis according to your concerns and discussion.

---

> > > > ### Comment · Reviewer_Y3bu · 2024-08-11
> > > >
> > > > I appreciate the authors' quick and convincing response. The empirical evaluation looks much more comprehensive now, which demonstrates the optimality of the proposed combinations and the sacrifice (i.e., compared to diffusion models) we need to make for the interpretability. My concerns are well-addressed and I will increase my scores.

---

### Official Review · Reviewer_SQvd · 2024-07-08

**Soundness:** 3
**Presentation:** 2
**Contribution:** 3
**Rating:** 6
**Confidence:** 4

**Summary:**

Researchers introduced a bidirectional weighted graph-based framework to explore factorized attributes and their interrelations within complex data. They proposed a -VAE module for extracting initial factors and utilized a multimodal large language model (MLLM) to uncover latent correlations and update weighted edges. Integrating these modules enabled their model to achieve superior unsupervised disentanglement and reconstruction performance, inheriting interpretability and generalizability from MLLMs.

**Strengths:**

1. The paper introduces a graph-based approach to model interrelationships within complex data, aiming to integrate background knowledge into Deep Reinforcement Learning (DRL). I find this idea novel and intriguing.
2. The paper presents a rigorous framework with clear exposition and straightforward methodology, making it accessible and easy to understand.

**Weaknesses:**

1. The paper lacks significant innovation at the neural network and algorithmic levels. While the introduction of DisGraph and its optimization methods is proposed to be effective, insufficient explanation is provided regarding why they work. Strengthening this aspect of the description would enhance the paper's persuasiveness.
2. The paper could provide a brief explanation of some methods used, such as the Somers' D algorithm, even if included in an appendix. Currently, this aspect appears somewhat incomplete.
3. An explanation of the update mechanism for the entire model at the end of Section 3 would be beneficial.

**Questions:**

1. How is the effectiveness of DisGraph ensured? What problems could arise if errors are introduced into DisGraph?

**Limitations:**

Yes

---

> ### Author Rebuttal · Authors · 2024-08-07
>
> *We greatly appreciate all of your valuable suggestions, which play a pivotal role in enhancing the quality of our paper. Below we address all your concerns.*
>
> **W1&W3: Detailed explanations of the optimization mechanism for the entire model**
>
> **R1:** Thanks for your feedback. We promise to enhance our manuscript with more detailed explanations of the model's optimization mechanism. Below, we offer a concise explanation of the optimization mechanism for clarity:
>
> During the training, the optimizable parameters of the encoder $E_{sem}$, DisGraph $G$ and decoder $D_{rec}$ are denoted as $\phi$, $\gamma$ and $\theta$, respectively. And the optimization objectives can be formulated as follows:
>
> $L_{gem}$ ($\phi$, $\gamma$, $\theta$) = $D_{\mathrm{KL}}(q_{\phi}(x, z)$,$p_{\gamma,\theta}(x, z))$,  $L_{dis} = -D_{\mathrm{KL}}\left(q_{\phi}(\mathbf{z}|\mathbf{x})  \||  p_{\theta}(\mathbf{z})\right) $
>
> $L_{total} = \lambda_{gem} L_{gem} + \lambda_{dis} L_{dis} + \lambda_{adv} L_{adv} $
>
> where the specific optimization processes can be formulated as:
>
> $\nabla_{\theta} L_{gem}(\phi,\gamma,\theta) \overset{x=D_{\theta}(z)}{=} -E_{z\sim q(z)}\nabla_{x}\left[\log\left(\frac{p_{\theta,\gamma}(x,z)}{ q_\phi(x, z)}\right)\right]\nabla_{\theta}x$
>
> $\nabla_{\phi} L_{gem}(\phi,\gamma,\theta) \overset{z=E_{\phi}(x)}{=} E_{x\sim p(x)}\nabla_{z}\left[\log\left(\frac{p_{\theta,\gamma}(x,z)}{ q_\phi(x, z)}\right)\right]\nabla_{\phi}z$
>
> $\nabla_{\gamma} L_{gem}(\phi,\gamma,\theta) \overset{z=G_{\gamma}(z_{dis})}{=} E_{x\sim p(x)}\nabla_{z}\left[\log\left(\frac{p_{\theta,\gamma}(x,z)}{ q_\phi(x, z)}\right)\right]\nabla_{\gamma}z$
>
> Furthermore, the optimization objective of the discriminator can be expressed as:
>
> $L_{adv} = L(D) = \frac{1}{N} \left[ \sum\limits_{i=0; (x_i, z_i) \in E_\phi}^{N} \text{softplus}(-D(x_i, z_i)) + \sum\limits_{i=0; (x_i, z_i) \in D_\theta}^{N} \text{softplus}(D(x_i, z_i)) \right]$
>
> where $\(D^*(x,z)=\log\left(\frac{p_{\gamma, \theta}(x,z)}{q_{\phi}(x,z)}\right)\)$. The discriminator $D$ can be used to fit $D^*$ to achieve gradient estimation and complete the training. You may also check the **R1and R3 for Reviewer d3xR**, for insights into the framework's definitions and workflow.
>
> **W2: A brief explanation of some methods used, such as Somers' D algorithm, will be beneficial**
>
> **R2:** Thanks for your suggestions. We will meticulously integrate comprehensive definitions and clarifications of all employed functions. Specifically, for the Somers' D algorithm you referenced, we provide the subsequent example to enhance your understanding. Suppose we have the sample dataset $S=\{(1,2),(3,1),(2,3)\}$:
>
> | **Variable** | Pairs                          | Value |
> |--------------|--------------------------------|-------|
> | $N_c$        | (1,2) vs (2,3)                 | 1     |
> | $N_d$        | (1,2) vs (3,1) and (2,3) vs (3,1) | 2     |
> | $T_y$        | None                           | 0     |
>
> Somers' D indicator can be calculated as follows:
>
> $ D =\frac{N_c-N_d}{N_c+N_d+T_y} = \frac{1 - 2}{1 + 2 + 0}  = -\frac{1}{3} $
>
> This obtains a value of approximately -0.33, signifying a negative correlation between variables $X$ and $Y$. This calculation demonstrates that the Somers' D metric is straightforward to calculate and is particularly applicable to ordinal variables. Furthermore, Somers' D is asymmetric and capable of distinguishing bidirectional relationships between variables. These characteristics make it highly suitable for integration into our model.
>
> **Q1: How is the effectiveness of DisGraph ensured? What problems could arise if errors are introduced into DisGraph?**
>
> **R3:** To ensure the effectiveness of DisGraph, we perform an ablation experiment by disabling the DisGraph (see Figure 7 in Section 4.5). Since completely removing DisGraph is infeasible due to its role in embedding bidirectional weighted relations, we alternatively utilize an initial version of DisGraph without the updating module $E_{gnn}$. Refer to the top-left in Figure 7, where the model, using an initial Graph, exhibits weakened or inaccurate relational awareness (e.g., the relation between *"Bald"* and *"Gender"* weakens). This observation demonstrates the effectiveness of DisGraph.
>
> Regarding your second question, to our knowledge, the errors in interrelation identification from MLLMs may lead to counterfactual outcomes. For example, applied with a less capable MLLM that does not recognize the positive correlations between *"banana"* and *"yellow"* or *"age"* and *"white hair"*, the model may obtain *a red banana* or *a white hair baby* learned from the errors introduced into DisGraph (refer to the counterfactual results by a weaker MLLM in **Figure A3 of the attachment**). In our perspective, even advanced MLLMs (e.g., GPT-4o) can exhibit a certain bias. To enhance robustness of GEM, we are investigating potential improvements both within the MLLM (e.g., data pre-processing, bias-aware modules, and knowledge editing) and externally (e.g., weight redistribution, debiasing modelling, and modifications to decoding strategies).

---

> > ### Comment · Reviewer_SQvd · 2024-08-10
> >
> > Thank you to the author for providing a thorough response to my question. I believe it has addressed my concerns regarding DisGraph. I will raise the score I have given and consider the paper to be an excellent one.

---

### Official Review · Reviewer_d3xR · 2024-07-08

**Soundness:** 3
**Presentation:** 2
**Contribution:** 3
**Rating:** 6
**Confidence:** 3

**Summary:**

To achieve fine-grained, interpretable and unsupervised disentangled representation learning (DRL), this paper proposes a new framework by integrating $\beta$-variational autoencoder ( $\beta$-VAE), multimodal large language model (MLLM) and graph learning into a single pipeline. Experimental results show that the proposed framework can achieve a better trade-off in the capability of disentanglement and quality of reconstruction than the evaluated baselines under different datasets.

**Strengths:**

The strengths of the paper are listed as follows.
1. The paper is well-motivated. Figure 1 clearly illustrates the limitations of the existing works in DRL and the advantages of the proposed framework, which clearly shows the motivation of the paper.
2. This paper provides a thorough and insightful summarization of the related work in DRL, which highlights the contribution of the proposed framework.
3. It is great that the authors could provide detailed and diverse qualitative results in the section of experiments. It can give the readers a more clear view to understand the effect brought by the proposed framework.

**Weaknesses:**

The weaknesses of the paper are listed as follows.
1. It would be better if the authors could first formulate the problem as an optimization problem mathematically before introducing the details of the method to solve the problem. In the problem formulation, the input, output, constraints and objectives should be clearly defined.
2. In the proposed framework, the input to the decoder $D_{rec}$ is not from the normal distribution but from the variable extracted from DisGraph. It is a critical point that needs to be highlighted in Figure 2. However, it is missing in Figure 2.
3. Some technical details are not clear as listed below.
A. The input to the decoder $D_{rec}$ is not from the normal distribution but from the variable extracted from DisGraph. But how to make this process differentiable for end-to-end training is unclear.
B. It is not clear how to train the graph learner in an unsupervised way. What is the loss function? What is the dimension of the node feature? Moreover, how to update the adjacency matrix of the DisGraph given the updated weights of the GNN?
C. In Figure 2, it is not clear about the usage of a set of extra images input to the landmark function.
4. It would be better if the authors could use some metric to quantify the disentanglement capability of the DRL algorithms and show the quantitative results to evaluate the proposed framework and the baselines.
5. It would be better if the authors could provide a more intuitive explanation of the loss function they designed. What is the usage of each term within the loss function? Why can they address the limitations of the existing challenges of DRL?

**Questions:**

The questions of the paper are listed as follows.
1. Is Equation (2) instead of Equation (1) the fundamental objective of vanilla VAE?
2. What are the differences among $p_{\theta}(z|x)$, $p_{\theta}(x|z)$ and $p_{\theta}(z)$? Why can they share the same parameters $\theta$?
3. Why can the MLLM have the capability to provide accurate attribute scores given the multimodal input provided by the framework?
4. If it has been proved that unsupervised DRL is impossible without extra prior, why does the proposed unsupervised framework work? What is the extra prior or inductive bias here?
5. Could you please give more explanation about why is the proposed algorithm to obtain the impact scores between the pair of attributes reasonable?
6. What are the potential use cases of the disentanglement capability from DRL?

**Limitations:**

The limitations are shown in the section of Weaknesses and Questions.

---

> ### Author Rebuttal · Authors · 2024-08-07
>
> *We value your insightful feedback and will refine our manuscript accordingly. Here, we address each of your concerns.*
>
> **W1: Detailed definitions of the model**
>
> **R1:** Thanks for your suggestions. We will detail the model's parameters, definitions, and optimization strategies in revision. Here, we present a brief overview of for clarity:
>
> | Component        | Input                                      | Output                                   |
> |------------------|--------------------------------------------|------------------------------------------|
> | Encoder $E_{sem}$ | Image $x \in \mathbb{R}^{n \times n}$      | Disentangled latent variable $z_{dis} \in \mathbb{R}^{pre}$ |
> | DisGraph $G$      | Disentangled latent variable $z_{dis} \in \mathbb{R}^{pre}$ | Correlation-involved latent variable $z \in \mathbb{R}^{pre}$ |
> | Decoder $D_{rec}$ |Correlation-involved latent variable $z \in \mathbb{R}^{pre}$   | Reconstructed image $\hat{x} \in \mathbb{R}^{n \times n}$ |
>
> where $pre$ is the pre-defined dimensionality for all latent variables. During training, the optimizable parameters of the encoder $E_{sem}$, DisGraph $G$ and decoder $D_{rec}$ are denoted as $\phi$, $\gamma$ and $\theta$, respectively. And the optimization objectives can be formulated as follows:
>
> $L_{gem}$ ($\phi$, $\gamma$, $\theta$) = $D_{\mathrm{KL}}(q_{\phi}(x, z)$,$p_{\gamma,\theta}(x, z))$,  $L_{dis} = -D_{\mathrm{KL}}\left(q_{\phi}(\mathbf{z}|\mathbf{x})  \||  p_{\theta}(\mathbf{z})\right) $
>
> $L_{adv} = L(D) = \frac{1}{N} \left[ \sum\limits_{i=0; (x_i, z_i) \in E_\phi}^{N} \text{softplus}(-D(x_i, z_i)) + \sum\limits_{i=0; (x_i, z_i) \in D_\theta}^{N} \text{softplus}(D(x_i, z_i)) \right]$
>
> $L_{total} = \lambda_{gem} L_{gem} + \lambda_{dis} L_{dis} + \lambda_{adv} L_{adv} $
>
> In the revised total loss, as distinct from the manuscript's version, we partition the original $L_{dis}$ into two components: $L_{gem}$ for reconstruction and $L_{dis}$ for disentanglement.
>
> **W2: Missing details in Figure 2**
>
> **R2:** Thanks for your thorough review. We have updated Figure 2 to highlight the input distribution of $D_{rec}$ accordingly.
>
> **W3_A: How to make it differentiable for end-to-end training in the model**
>
> **R3:** Based on the latent variable $z_{dis}$ from encoder $E_{sem}$, we generate $n$ nodes, each representing one of the $n$ semantic attributes. In practice, each node mirrors $z_{dis}$'s dimensionality but isolates the $i$-th attribute by masking all dimensions except the $i$-th, thereby to learn the $i$-th attribute. These nodes, combined with interrelations from $P_{rel}$, form the DisGraph, which outputs the embedding matrix $T$. This matrix is utilized to calculate $z$ by averaging each volume, subsequently decoded by $D_{rec}$ to reconstruct the image with our loss functions. All aforementioned modules are differentiable.
>
> **W3_B: Unclearness of the Graph learner training**
>
> **R4:** As described in Line 195, we follow the instructions by Liu et al. to unsupervisedly train the graph learner. Specifically, the adjacency matrix are updated within $E_{gnn}$, by the Structure Bootstrapping Mechanism and Multi-view Graph Contrastive Learning.
>
> **W3_C: Unclearness in Figure 2 about Landmark**
>
> **R5:** In this work, we only apply off-the-shelf landmark models for data pre-processing, which does not require any extra data in the databases. We will accordingly clarify it in Figure 2.
>
> **W4: Disentanglement metrics**
>
> **R6:** Our research emphasizes incorporating attribute interrelations, where enhancing disentanglement is not our primary objective (refer to **R4 to Reviewer qyA4**). However, to address your concern, we have conducted extra experiments as shown in **Figure A4 in the attachment**.
>
> **W5:  Explanations of the loss function**
>
> **R7:** We have detailed our loss functions in **R1**. Regarding your final concern, our approach addresses DRL’s limitations by: 1) contributing to a more practical paradigm; 2) balancing reconstruction and disentanglement abilities by adjusting $\lambda_{gem}$ and $\lambda_{dis}$.
>
> **Q1:  Is Equation (2) the fundamental objective of vanilla VAE?**
>
> **R8:** Equation (2) represents the objective function of the $\beta$-VAE, while Equation (1) represents the likelihood estimation for vanilla VAE.
>
> **Q2:  Justifications of parameters $\theta$**
>
> **R9:** We adopt the original VAE notations: $p_{\theta}(z)$ implies $z$ follows a standard normal distribution without parameters. In $p_{\theta}(z|x)$ and $p_{\theta}(x|z)$, $\theta$ denotes decoder parameters. $p_{\theta}(z|x)$ is used solely for theoretical derivation of the variational lower bound, with no parameter sharing in practice.
>
> **Q3:  Why can MLLMs provide accurate scores?**
>
> **R10:** Theoretically, the impressive capabilities of MLLMs, especially in realistic AI generation, reveal that they can comprehend real-world concepts at a certain level, informed by in-depth studies (see Section 2.3); Experimentally, our evaluations of SOTA MLLMs confirm their reliability (see Section 4.4 and **Figure A1 in attachment**).
>
> **Q4:  Reliance on unsupervised DRL**
>
> **R11:** Refer to the **R2 for Reviewer qyA4**, the DRL branch extracts and initializes latent factors. Although these factors exhibit partial independence, this limitation is acceptable as their interrelations will be refined and updated by the MLLM branch and DisGraph. Hence, our model does not require any external supervision or priors.
>
> **Q5: Explanations on interrelations determining**
>
> **R12:** For determining the interrelationships between two attribute scores from MLLMs, we employ correlation analysis algorithms (e.g., Somers' D). Due to spatial constraints, please refer to **R2 for Reviewer SQvd** for the details of this part.
>
> **Q6: Potential applications**
>
> **R13:** Our practical and relation-aware DRL framework can be applied in domains like AI-generated content, explainable AI, medical imaging, robotics, and autonomous vehicles.

---

> > ### Comment · Reviewer_d3xR · 2024-08-13
> >
> > Thanks for the detailed rebuttal from the authors. I think it solved most of my concerns. Given the feedback, I think a thorough revision should be made for the final publication. So please remember to make the revisions you promised for the final version of the paper. I have raised my score for this paper.

---

### Official Review · Reviewer_qyA4 · 2024-07-11

**Soundness:** 3
**Presentation:** 2
**Contribution:** 3
**Rating:** 5
**Confidence:** 4

**Summary:**

The paper presents a novel framework that integrates β-VAE with multimodal large language models within a graph structure, DisGraph, to enhance disentangled representation learning. This approach allows for effective handling of complex and interdependent data attributes in an unsupervised manner. The model dynamically updates relationships between attributes, improving disentanglement and interpretability compared to traditional methods. Extensive experiments demonstrate its superior performance in both disentanglement and reconstruction.

**Strengths:**

1. The paper successfully integrates β-VAE with multimodal large language models (MLLMs) within a graph-based framework, which is a novel approach.
2. The use of a graph-based approach to model the relationships among attributes addresses a significant gap in existing DRL methods.
3. The experiments are well-designed, covering various aspects of the model’s performance.

**Weaknesses:**

1. The paper would benefit significantly from clearer writing and better organization. The paper occasionally uses technical terms and concepts without adequate definitions or explanations.
2. While the paper addresses the unrealistic assumption of statistical independence in many DRL methods, the solution proposed still relies heavily on the disentanglement abilities of β-VAE, which itself often presupposes some level of independence or weak dependence among latent variables.
3. Also, the reliance on pre-trained multimodal large language models might introduce biases from these models into the disentangled representations.
4. A deeper theoretical analysis of why and how the inclusion of interrelations leads to better disentanglement would be valuable.

**Questions:**

1. How are the generated node embeddings T generated by DisGraph associated with the losses introduced in Section 3.1?

**Limitations:**

Yes

---

> ### Author Rebuttal · Authors · 2024-08-07
>
> *We greatly appreciate your insightful comments and commit to refining our manuscript based on your suggestions. Below, we address all your concerns.*
>
> **W1: The paper would benefit significantly from clearer writing, organization and explanations**
>
> **R1:** Thanks for your suggestion, we promise to enhance the manuscript on the writing style and structural organization. In addition, we will ensure that all technical concepts presented in the paper are associated with clear and comprehensive definitions or derivations (e.g., Somers’ D algorithm).
>
> **W2:  The heavy reliance on the disentanglement abilities of β-VAE, which itself often presupposes some level of independence or weak dependence.**
>
> **R2:** Thanks for your insightful comments. We wish to re-emphasize the key of our work: To move beyond the unrealistic independence assumption of traditional DRL, we propose a more **logical and practical DRL paradigm** that involves the bidirectional interrelations between attributes. In this framework, **$\beta$-VAE branch is only employed to extract and initialize latent factors.** The weak dependence or partial independence among initial factors is acceptable, as their interrelations will be subsequently determined, overwritten, and updated by the proposed MLLM branch and DisGraph. In this way,  our method aligns more closely with real-world dynamics and offers broader applicability.
>
> **W3:  Reliance on pre-trained multimodal large language models might introduce biases**
>
> **R3:** Our framework is based on the belief that MLLMs, including their future iterations, are sufficiently robust to comprehend logical rules of reality (e.g., aging brings wrinkles, sunrise brings light, etc.). These rules are represented as interrelations between entities. On this basis, we have corroborated the efficacy of MLLMs using ground truths to affirm their effectiveness in straightforward scenarios (see **Section 4.4 and Figure A1 in the attachment**). However, we totally agree with you that even the most powerful MLLMs (e.g., GPT-4o) can exhibit bias on limited pre-training data. To address this, we are exploring potential solutions from both within the MLLM (e.g., data pre-processing, bias-aware module, knowledge editing, and etc.) and from our end (e.g., weight redistribution, debiasing modelling, decoding strategy modification and etc.).
>
> **W4:  A deeper theoretical analysis of why and how the inclusion of interrelations leads to better disentanglement would be valuable**
>
> **R4:** Thanks. It is really meaningful and interesting to discuss **"what is a better disentanglement"**. If it means the better performance on independently decomposing factors, then the inclusion of interrelations might not seem beneficial; however, if it refers to a better performance/practicality for real and complex scenarios, our disentanglement paradigm excels by statistically capturing the logical rules of real world.  Specifically, the inclusion of interrelations can be beneficial in model generalizability, counterfactual reasoning and practical usages. We will further conduct analysis and discussion on this point in the manuscript, through both theoretical analysis and experimental investigations.
>
> **Q1:  How are the generated node embeddings T generated by DisGraph associated with the losses?**
>
> **R5:** Our apologies for the unclear descriptions. In brief, DisGraph generates the embedding matrix $T$ based on updated parameters to derive the latent variable $z$ through the aggregation of node embeddings. $z$ is subsequently decoded by $D_{rec}$ to reconstruct the image within the bounds of our loss functions. We welcome your reference to the **R1 for Reviewer d3xR** for a comprehensive discussion of the model's loss functions and workflow.

---

> ### Author Response · Authors · 2024-08-12
> **Hope for your further comments**
>
> Thank you for your continued efforts. We have provided comprehensive rebuttals and tried to address the concerns raised in your reviews. Please take the time to review if possible, if you have any further questions or require additional clarification, please let us know and we welcome discussions in any format . Thanks again.

---

> > ### Comment · Reviewer_qyA4 · 2024-08-13
> > **Official Comment by Reviewer qyA4**
> >
> > I thank the authors for their responses. I will raise my score.

---

### Official Review · Reviewer_nREj · 2024-07-12

**Soundness:** 3
**Presentation:** 3
**Contribution:** 3
**Rating:** 6
**Confidence:** 3

**Summary:**

Objective
The paper presents a novel framework for disentangled representation learning, which aims to identify and separate the underlying factors of variation in complex data. The primary goal is to enhance the interpretability and robustness of data perception and generation models.

Methodology
The proposed method integrates a β-VAE (Variational Autoencoder) with a bidirectional weighted graph framework and utilizes Multimodal Large Language Models (MLLM) to discover and rank potential relationships between factors. This integrated approach ensures fine-grained, practical unsupervised disentanglement.

Contributions
1. Innovative Framework: Combining β-VAE with MLLM and a bidirectional weighted graph for enhanced disentangled representation learning.
2. Fine-Grained Disentanglement: Achieves more detailed and practical separation of data factors.
3. Improved Interpretability: The model inherits the interpretability and generalization capabilities of MLLMs.
4. Robust Evaluation: Shows superior performance in terms of disentanglement and reconstruction quality on benchmark datasets.

**Strengths:**

Originality.
The paper presents a unique combination of β-VAE and Multimodal Large Language Models (MLLM) within a bidirectional weighted graph framework. This novel integration allows for capturing and ranking complex relationships between factors, addressing limitations in previous methods.

Quality.
The methodology is rigorous, with well-designed components and thorough evaluations on the CelebA and LSUN datasets. The use of Graph Neural Networks (GNNs) for optimizing the DisGraph demonstrates a high level of sophistication and effectiveness.

Clarity.
The paper is well-organized and clearly written, with detailed explanations and helpful diagrams. Each component of the framework is logically explained, making the complex methodology accessible to readers.

Significance.
The framework enhances interpretability and robustness in disentangled representation learning, with practical implications for image generation and AI explainability. It sets a new direction for future research.

**Weaknesses:**

1. Dependence on MLLM: The framework heavily relies on Multimodal Large Language Models (GPT-4) to discover and rank relationships between factors. This dependence can be problematic if the MLLM is not sufficiently trained on relevant data or if it introduces biases present in its training corpus. Additionally, relying on a single score from MLLM may not be sufficiently convincing, making this step overly dependent on the accuracy and reliability of MLLM. Despite the evaluations in section 4.4, which assess the accuracy of the scores, having specific guidelines or principles for MLLM scoring would be more convincing than relying solely on a single score.

2. Lack of Detailed Attribution: While the paper introduces a novel framework, it lacks detailed explanations on how each latent representation specifically maps to distinct attributes. This can make it challenging to understand and interpret the exact role of each variable in practical applications.

**Questions:**

1. I did not fully understand how the paper finds the correspondence between the latent representations output by the Encoder and the semantic attributes. For example, in Figure 1, how do the latent representations obtained through the Encoder correspond to attributes such as hat, eyes, etc.? Could you provide a detailed explanation of this process?

2. In the step where MLLMs are used to evaluate attributes (Figure 3), is there a specific principle guiding the MLLM to score the attributes? For instance, what is the difference when the MLLM scores the same attribute as 2, 3, or 4? Additionally, what would the results be if other MLLMs mentioned in your related work section were used for scoring instead of relying solely on the GPT-4 series?

**Limitations:**

Yes

---

> ### Author Rebuttal · Authors · 2024-08-07
>
> *We greatly appreciate your insightful comments and commit to refining our manuscript based on your suggestions. Below, we address all your concerns.*
>
> **W1: Dependence on MLLM**
>
> **R1:** Thanks for the insightful comments. First, we would like to clarify that our model leverages the commonsense knowledge embedded in MLLMs to predict interrelations. This is predicated on the assumption that MLLMs, including their future iterations, are powerful and reliable enough to comprehend the physical rules of the real world (e.g., aging brings wrinkles, sunrise brings light, etc.). To substantiate the reliability, we evaluate the scores from SOTA MLLMs with ground truths, as detailed in **Section 4.4 and Figure A1 in the attachment**.
>
> Nevertheless, we totally agree with your comments, that a solitary score is not convincing enough to represent MLLMs' knowledge. Towards it, two perspectives of considerations are undertaken: 1) enhancing the prompting guidelines to have more informative outputs (e.g., likelihood probabilities, multi-level categorical outputs, averaged iterative scores and etc.); 2) interpreting lightweight bias mitigation modules with certain prior knowledge (e.g., [1] and [2]).
>
> *[1]Hauzenberger et al., Modular and on-demand bias mitigation with attribute-removal subnetworks, 2023.*
>
> *[2]Kumar et al., Parameter-efficient modularised bias mitigation via AdapterFusion, 2023.*
>
> **W2: Lack of Detailed Attribution**
>
> **R2:** Thanks. Actually, it is a frequent question why unsupervised DRL models possess the ability to align specific attributes. We provide a brief discussion here and will offer a comprehensive explanation in the revision.
>
> To address your concern, we need to analyze it from the information-theoretic perspective. As per the Information Bottleneck (IB) theory [3], constraining the information input to the DRL model (e.g., by the penalty coefficient $\beta$ in $\beta$-VAE), inherently enables the model to **identify and learn the most representative factors for successful reconstruction.** For instance, when trained on the Shapes3D (a collection of simple, synthetic objects) with a merely three-dimensional latent variable, the model tends to learn the most critical factors which are observed to be *"object colour"*, *"object shape"*, and *"background shape"*. **These attributes are aligned in the three dimensions, organized in order of their contribution to reconstruction.** Similarly for facial images, the model spontaneously learns and organizes the most informative attributes (e.g., hair, gender and etc.) in the disentangled dimensions.
>
> In addition, in our framework, we employ a landmark detection for pre-processing, extracting pivotal object features and discarding image redundancies via cropping and resizing. It further helps the mapping of each latent representation to unique attributes.
>
> *[3] Burgess et al., Understanding disentangling in $\beta$-VAE, 2018.*
>
> **Q1: I did not fully understand how the paper finds the correspondence between the latent representations output by the Encoder and the semantic attributes**
>
> **R3:** In the response for W2, we have clarified that for a collection of data, the most informative attributes will be unsupervisedly learned by the $\beta$-VAE and sequentially aligned across several dimensions. It highlights the role of $\beta$-VAE branch to **disentangle and initialize the attributes**. Nevertheless, these initialized attributes need a further human observation to extract textual concepts for subsequent MLLM prompting.
>
> **Q2: Is there a specific principle guiding the MLLM to score the attributes? For instance, what is the difference when the MLLM scores the same attribute as 2, 3, or 4? What would the results be if other MLLMs were used for scoring?**
>
> **R4:** In the paper, Figure 3 illustrates the specific guiding principle, which prompts MLLMs to classify one attribute into multiple degrees on its own judgement. Specifically, it assigns scores ranging from 0 to 5 for each attribute, where 0 indicates the attribute’s absence, and 5 denotes its highest expression. For example, for scoring attribute *''smile"*, MLLMs tend to assign a score of 0 for the absence of a smile and a score of 5 for a full laugh, **according to the statistical distribution they have learned from extensive data.**
>
> Towards your 2nd question, we wish to further clarify: since the goal of the MLLM branch in GEM is to discover the interrelations, **the statistical relativity between two attributes is of primary concern, rather than the absolute scores for the individual attribute (see Figure A2 in the attachment).** For example, given a collection of facial images, it is acceptable if the scores of *"age"* and *"bald"* exhibit a positive correlation, even if the specific score values are fluctuating.
>
> Addressing your final concern, we believe that a less capable MLLM would hinder the learning of attribute interrelations. We substantiated this by replacing GPT-4o with the inferior GLM and assessing the outcome. **Figure A3** in the attachment demonstrates that GLMs' limited perceptual capacity yielded counterfactual outcomes. Therefore, the MLLM bias mitigation methods mentioned in **R1** are valuable.

---

> > ### Comment · Reviewer_nREj · 2024-08-12
> >
> > In general, I am satistifed with the author feedback. As reflected in my score, the paper has its merits, and is above the acceptance bar in my opinion.

---

### Author Rebuttal · Authors · 2024-08-07

*Dear reviewers,*

*First of all, we would like to thank all reviewers for their time and efforts in reviewing this paper. These insightful comments are really helpful in guiding to improve the manuscript.*

***We have made our efforts to meticulously address each concern raised by the reviewers. Please refer to separate responses for details. We have also attached a one-page PDF to support our responses.***

*We hope that the responses sufficiently address the reviewers' concerns. And we are open to further discussions if there be any unresolved issues.*

*Finally, we promise a careful revision of the manuscript according to these comments and discussions. We have released our code online (as attaching external links is not allowed, we have sent an anonymous GitHub link to the AC as required), and hope our work can provide valuable insights to the community.*

*Sincerely yours,*

*Authors*

---

### Decision · Program_Chairs · 2024-09-25

**Decision:**

Accept (poster)

**Comment:**

Reviewers noted the novel approach combining β-VAE with MLLM and a bidirectional weighted graph for disentangled representation learning, as well as enhanced interpretability, as the paper's principal strengths. Initially, the reviewers raised a list of concerns, notably that the paper occasionally lacks adequate definitions or explanations for technical terms and concepts, lacks a deeper theoretical analysis of why and how the inclusion of the interrelations leads to better disentanglement, lacks detailed explanations,  clear definitions for constraints and objectives,  lacks metrics to quantify the disentanglement capability of the DRL algorithm,  an explanation for the designed loss function, contains unclearness in the graph learner training, and an unclear description of how the node embeddings are associated with the losses. Additionally, there is a lack of comparison with more advanced generative models, such as diffusion models.

The authors' rebuttal and the discussion that followed have elucidated many of these concerns, and the authors have promised to include the proposed improvements in the final version of the paper. Based on this, acceptance is recommended. The authors are asked to thoroughly follow up with the requested adjustments, with special attention to the following points: 1) explain why such generalized disentangled representations, despite not fulfilling the principal standard requirement of having independent factors of variations, may nevertheless be useful in practical situations; 2) add more evaluations with disentanglement metrics such as DCI, FactorVAE, and SAP;  3) it would be beneficial to provide more experimental evidence of the advantages of the paper's approach on another challenging dataset.